# Mapping the Auditory Space of *Culex pipiens* Female Mosquitoes in 3D

**DOI:** 10.3390/insects14090743

**Published:** 2023-09-04

**Authors:** Dmitry N. Lapshin, Dmitry D. Vorontsov

**Affiliations:** 1Institute for Information Transmission Problems of the Russian Academy of Sciences, Bolshoy Karetny per. 19, 127994 Moscow, Russia; lapshin@iitp.ru; 2Koltzov Institute of Developmental Biology Russian Academy of Sciences, Vavilova 26, 119334 Moscow, Russia

**Keywords:** directional hearing, mosquito, *Culex pipiens*, flagellar auditory receiver, particle velocity

## Abstract

**Simple Summary:**

Mosquitoes possess one of the best-developed and sensitive hearing systems among insects. Their auditory Johnston’s organs located at the antennae bases include several thousand radially distributed sensory cells. Male mosquitoes use their hearing for acoustic courtship behavior, while the function of hearing in blood-sucking female mosquitoes is poorly studied. In addition to courtship behavior, hearing is presumed to be used for host detection, including the use of human voices as an attraction cue. Since mosquitoes spread dangerous diseases such as West Nile fever, understanding their hearing system is of crucial importance. We studied the auditory system of *Culex pipiens* female mosquitoes using behavioral and electrophysiological experiments and created a three-dimensional model of the mosquito auditory space. The in-flight position of antennae was found optimal for binaural hearing focused primarily in front of, above and below a mosquito. By varying the antennae position a mosquito can adjust the directional properties of hearing depending on behavioral context. According to our findings, the auditory system of female mosquitoes has enough resolution to estimate the direction to the sound source, while its frequency range enables detection of sounds produced by other flying mosquitoes and human hosts.

**Abstract:**

The task of directional hearing faces most animals that possess ears. They approach this task in different ways, but a common trait is the use of binaural cues to find the direction to the source of sound. In insects, the task is further complicated by their small size and, hence, minute temporal and level differences between two ears. A single symmetric flagellar particle velocity receiver, such as the antenna of a mosquito, should not be able to discriminate between the two opposite directions along the vector of the sound wave. Paired antennae of mosquitoes presume the usage of binaural hearing, but its mechanisms are expected to be significantly different from the ones typical for the pressure receivers. However, the directionality of flagellar auditory organs has received little attention. Here, we measured the in-flight orientation of antennae in female *Culex pipiens pipiens* mosquitoes and obtained a detailed physiological mapping of the Johnston’s organ directionality at the level of individual sensory units. By combining these data, we created a three-dimensional model of the mosquito’s auditory space. The orientation of the antennae was found to be coordinated with the neuronal asymmetry of the Johnston’s organs to maintain a uniformly shaped auditory space, symmetric relative to a flying mosquito. The overlap of the directional characteristics of the left and right sensory units was found to be optimal for binaural hearing focused primarily in front of, above and below a flying mosquito.

## 1. Introduction

Sound localization is one of the major tasks of the auditory system. In animals, it is differently solved depending on which components of the sound wave, pressure or particle velocity are being detected [1]. Hearing systems based on pressure receivers (tympanal ears) use binaural cues such as time and level differences to localize sounds [2]. However, the amplitude difference at the two ears, as well as the timing difference may be very small even in mammals [3], not to mention tinier creatures such as insects. To solve this problem, two ears often become mechanically or acoustically connected, thus creating a pressure-difference receiver, which enables amplification of tiny acoustic cues into larger interaural differences, which can be detected by the nervous system [4,5,6].

On the other hand, particle velocity receivers, such as antennae of mosquitoes, are inherently directional [1,7,8]. The radially symmetrical structure of the Johnston’s organ (JO), a modified second segment of the insect antenna, means that only a fraction of the circular array of sensory cells are activated in response to a sound source. As the air particles move back and forth during the propagation of a sound wave, a single mosquito JO and other similarly designed auditory receivers should not be able to distinguish sounds that come from the two opposite directions. One may assume that each antenna is functionally omnidirectional and only the inter-antennal amplitude differences (IADs) are used to provide the localization cues, thus converging the functionality of particle velocity receivers to that of pressure receivers. Although the modelling experiments demonstrated that IADs in mosquito may be sufficient for the task of the sound localization [9], the main evidence in favour of a more complex mechanism of auditory directionality in mosquitoes is an extremely large number of sensory neurons in the JO [10], compared to insect tympanal ears [11].

Ablation of one antenna led to a reduced ability to locate a female [12], however, it did not completely prevent the auditory behavior. One may assume that the above-mentioned directional limitations of a single mosquito antenna could be overcome by the movement of a mosquito relative to the sound source. However, due to competition in a swarm, male mosquitoes must rapidly detect fast-flying females, which would require a mechanism of instant directionality.

A second antenna at a short distance from the first one and held parallel to it would provide little additional data on directionality. However, insects, and mosquitoes in particular, rarely hold their antennae in parallel, which opens up the possibility for additional interaural cues. The most obvious approach to instantly determine the direction to the sound source from the two opposite ones, provided by each of the two antennae, would be to compare the responses originating in the left and the right JOs based on the directional mapping in each of them. However, the binaural comparison may be rather complicated, as sound fields are usually strongly divergent close to small sound sources (such as a female mosquito), and bilateral particle velocity receivers may experience vastly different vector fields depending on the distance from the sound source [1].

Although the JOs are morphologically symmetrical, the responses of auditory neurons may not be physiologically symmetrical. In *Culex* male mosquitoes, it was demonstrated that two of the four quadrants in the JO contained more responding auditory cells [13]. Such asymmetry may speak in favor of the presence of ‘areas of interest’ in the three-dimensional auditory space of a mosquito and may indirectly point to the mechanisms of interaural comparison. However, functional asymmetry in the JO could arise from a possible experimental bias when recording the auditory responses from only a fraction of the antennal nerve. Such a bias is hard to avoid in experiments on male mosquitoes with their large extended antennae, which limit the direction of an electrode insertion.

The limitation can be overcome by recording from the JO of a female mosquito. The antennal fibrillae of female mosquitoes are smaller and allow for the insertion of a recording electrode from multiple directions, which would significantly decrease the possible bias caused by the systematic selection of recording sites within the antennal nerve.

The question ‘What do female mosquitoes hear?’ demands much attention by itself. The auditory system of biting female mosquitoes is poorly studied compared to that of the conspecific males, the outstanding listeners. Since mosquitoes spread dangerous diseases such as West Nile fever, the understanding of their hearing system is of crucial importance. Our very limited knowledge on such a practically important subject can be at least partially explained by the fact that for a long time it was difficult to demonstrate any behavioral auditory responses in female mosquitoes, although they also possess a sophisticated auditory system, only surpassed by their male counterparts.

Pre-copulatory acoustic interaction, demonstrated for a number of mosquito species [14,15,16,17,18], suggests that a female mosquito hears the flight tone of a male. Such hearing is possible if a female detects the distortion products of the nonlinear mixing of its own flight tone with that of a male [15,19,20,21,22]. However, there is evidence that previously reported “harmonic convergence” events are only a random by-product of the mosquitoes’ flight tone variance and not a signature of acoustic interaction between males and females [23].

Among bloodsucking dipterans, there are examples of distant attraction to the communication sounds of frogs in midges [24,25,26] and mosquitoes [27,28,29]. A study by Menda et al. [30] suggests that the *Aedes* female mosquitoes can be attracted to sound frequencies similar to those of human speech. Although the results on the female mosquito audition are far from conclusive, from the existing behavioral studies one can reason that the bloodsucking females of some dipterans, and mosquitoes in particular, can perceive the direction towards the sound source.

The lack of direct physiological data on auditory directionality in female mosquitoes, together with the above-mentioned experimental convenience, formed the basis of this work. From the previous comparative physiological study [31] we already knew that the auditory neurons in female mosquitoes are tuned to different frequencies. Accordingly, to make a correct measure of the auditory threshold of a given unit, first we measured its tuning frequency. Then, stimulated by the sound of that frequency, we quantified the directionality of the same unit by measuring the thresholds of its response to sounds coming from different directions. Such a procedure, repeated for a significant number of auditory units, gave us the overall directional characteristics of the JO and the estimate of the angular resolution of the auditory space. To test for the hypothesized irregularities in the directional distribution of the auditory units, we studied both left and right JOs.

On the next step, to analyze the auditory space of a mosquito possessing bilateral antennae, we complemented the individual auditory properties of each JO with the data on the relative in-flight orientation of the antennae. Towards this, we photographed flying mosquitoes, measured the orientations of their antennae in two planes and estimated the variation of these parameters. The combination of the antenna measurements with the dataset on the directionality of individual auditory units implied a three-dimensional representation of the data, which we achieved by designing a graphical computer model to present the results in a simplified and easy-to-grasp form.

In three dimensions, the neuronal irregularities mirrored in the left and right JOs were found to fit well with the in-flight orientation of the antennae as to jointly provide a uniformly shaped auditory space. Here, using the mosquito JO as a model system, we describe a common methodical framework to assess directionality in similarly organized bilateral particle velocity receivers.

As the final goal of this study, we explored if there are regions in the female mosquito auditory space to which it pays special attention in terms of binaural hearing. We believe that the answer to this question will lead to the design of more adequate behavioral experiments on female mosquitoes.

## 2. Materials and Methods

### 2.1. Animals

Females of the *Culex pipiens pipiens* L. were captured in the wild in the Moscow region of the Russian Federation. Experiments were performed at the Kropotovo biological station (54° 51′ 2″ N; 38° 20′ 58″ E) from August to September 2016–2021.

### 2.2. Behavioral Experiments

Imaging of freshly captured mosquitoes was performed in the field, as we noticed that mosquitoes kept in the laboratory for one or two hours flew continuously for significantly shorter periods. We placed individual mosquitoes into a circular plastic container, diameter 85 mm, depth 26 mm. Its flat walls were left transparent, while the curved side walls were covered with non-transparent film. The container was positioned either vertically (viewed from a side) or horizontally (viewed from above). In the former position, it was lit either by sunlight or by white LED array, intensity ca. 2000 lm, both through a light-scattering paper to provide a uniform bright background for a brightfield imaging. The other flat side of the container was attached to a tube, which in turn was connected to a photo camera with a macro lens (Olympus OM-D E-M10 II, 60 mm f/2.8). The pitch and roll of the camera were minimized relative to the horizon by means of a tripod. When viewed from above, the container was lit by scattered sunlight, and the same camera mounted on a tripod and facing down was used for imaging. Lateral and dorsal imaging for the same mosquito was done sequentially, one series after another. Experiments were performed at the end of August, from 1 p.m. to 6 p.m., substantially before the swarming hours of *Culex pipiens*. The mating status of the female mosquitoes was not checked.

Mosquitoes (n = 22) were serially photographed during the periods of flight, either spontaneous or caused by a startle from a shadow. From the images acquired in lateral view, we selected those where a mosquito was viewed strictly laterally (the second antenna was either close to or was hidden behind the first one) and was more than 1 cm from each wall. The selected set of images (three per each mosquito) was cropped and then the following angles were manually measured using the FIJI software package, version 1.52p [32]: antenna to horizon; abdomen to horizon; and between the two antennae (viewed from above, projection to the horizontal plane).

From the measurements in lateral view, we knew that the flying mosquito holds its antennae at an angle to the horizon. When viewed from above, this angle does not allow to directly measure the inter-antennal angle. One way to overcome this issue would be to photograph a mosquito in two projections simultaneously. However, as we required only the average value of the inter-antennal angle, we corrected the measured value using the averaged antenna-to-horizon angle according to the equation, derived from the geometry of the system:ξ = arctg(cos(β) ∙ tg(α/2)),(1)
where ξ is a half of the true inter-antennal angle, α is the measured inter-antennal angle, projected to the horizontal plane, and β is an angle between the antennal plane (a plane formed by the two antennae, Figure 1A, dashed blue line) and the horizon.

### 2.3. Microelectrode Recordings

The method for measuring the directional properties of the JO sensory units was reported in detail in our previous study on male mosquitoes [13].

Experiments were conducted in laboratory conditions with air temperature 17–24 °C. Focal extracellular recordings from the axons of the antennal nerve were made with glass microelectrodes (1B100F–4, WPI Inc., Sarasota, FL, USA) filled with 0.15 M sodium chloride and inserted at the scape–pedicel joint (Figure 2A). After the penetration of the cuticle, the electrodes had a resistance of 10–40 MΩ.

While penetrating the antennal nerve by the electrode, the preparation was continuously stimulated with tonal pulses (amplitude 60 dB SVPL, duration 80 ms, period 600 ms and dorso-ventral direction of acoustic vector (0°)). During this searching procedure the groups of the JO neurons situated orthogonal to the antenna oscillation could be overlooked, so the vector of the acoustic wave was periodically changed by 90°. The searching frequency was 100–130 Hz in the main series of experiments and 70 Hz during the search for low-frequency-tuned units (see below). We considered the recording site acceptable when the amplitude of response increased above 0.5 mV (peak to peak, Figure 2C).

In this study, a number of axons of primary sensory neurons (PSNs) contributed to the extracellular recording. Although we could not estimate this number in each recording, our previous studies [31,33], which included the intracellular recordings from the axons of PSNs, suggested that the number of simultaneously recorded axons was not large. In our experience, the stability of intracellular recording from the axon of the JO PSN is hard to maintain during the time required for the measurement of thresholds at different frequencies or directions of sound. For the sake of the stability of the recording, in this study we used only extracellular recording of neuronal responses. We use the terms ‘unit’ or ‘sensory unit’ in the sense of one or several axons belonging to the PSNs of the JO, closely located within the antennal nerve and sharing indistinguishable frequency and phasic properties, thus representing a single functional unit.

### 2.4. Acoustic Stimulation

We used two orthogonally oriented Scandinavia 75 (DLS, Örebro, Sweden) stationary speakers to create a vector superposition of acoustic waves at the point of mosquito antenna (Figure 2A,B), as described in detail in [13]. The mosquito was positioned at the crossing of the axes of two speakers in such a way that the antenna′s flagellum was perpendicular to the directions of sound waves originating from each of the two speakers (Figure 2B). This approach enabled us to set the desired direction of the acoustic vector relative to the antenna flagellum.

The moving parts of the speakers had a low resonant frequency (90 Hz). Due to the considerable response lag of the dome of the speaker and its support, the emission phase delay increased with the signal frequency up to the point of inversion. To stabilize the phase delay, a phase correction depending on the stimulation frequency was included in the speaker control circuit. Both speakers were covered by a metallic mesh to screen the recording electrode from the electrical signals that drove the speakers.

The sinusoidal stimuli were generated by the digital-to-analog converter LA-DACn10m1 (Rudnev-Shilyaev, Moscow, Russian Federation). Acoustic calibration of the stimulating device was performed with an NR-231-58-000 differential capacitor microphone (Knowles Electronics, USA) attached to a micropositioner with an axial rotation feature and set in the position of the mosquito. The same microphone placed 2 cm from the mosquito was used to record the stimulation signals during the experiments.

The speakers were powered by the home-made amplifier via a passive Sin–Cos (SC) transducer which produced two derived signals with the amplitudes
(2)$$A_{1}=\;0.25\cdot U\cdot \cos\left(\frac{\pi }{180}\cdot \left(\varphi +45\right)\right)$$
(3)$$A_{2}=\;0.25\cdot U\cdot \sin\left(\frac{\pi }{180}\cdot \left(\varphi +45\right)\right)$$
where *A*_1_ and *A*_2_ are the amplitudes of the signals for the first and the second speaker, respectively; U is the alternating voltage at the input of the SC transducer; φ is the angle between the dorso-ventral axis passing through the mosquito′s head and the vector of the vibrational velocity of the air particles. An increase in φ corresponds to the clockwise rotation of the velocity vector when viewed from the mosquito′s head along the antenna.

The resulting direction of the air vibration velocity in the stimulating system was determined by the vector superposition of the signals from both speakers. Changes in the sound wave direction relative to the mosquito in 15° steps were accomplished by coordinated switching of voltage dividers in the SC transducer. For those angles at which the values of the functions sin(φ + 45) or cos(φ + 45) were negative, the signal polarity was inverted by switching the terminals of the speakers.

The differential microphone together with its amplifier was previously calibrated in the far field using a B&K 2253 sound level meter with a B&K 4135 microphone (Brüel & Kjær, Nærum, Denmark). All sound level data in this study are given in the logarithmic scale in dB RMS SPVL (root mean square sound particle velocity level), with a reference level of 0 dB being equal to 4.85 × 10^−5^ mm/s, which corresponds in the far field to the standard reference sound pressure of 20 µPa.

The stereotyped way of positioning the mosquito relative to the experimental setup, especially to the recording electrode, could lead to selective recording only from a certain part of the antennal nerve. To minimize the probability of such a bias, the mosquito was occasionally positioned ventral side up (180° rotation, n = 15) or rotated by ±45° (n = 16) or ±90° (n = 37) around the longitudinal axis of its body. In each case, the recording electrode was inserted through a different part of the scape–pedicel joint. The directional data measured in all such experiments were corrected accordingly. To make an additional control for the above-mentioned experimental bias, in a separate series of experiments (n = 24) we performed recordings from the right JO instead of the left one, keeping similar all other properties of the experimental setup. To avoid confusion due to the possible symmetrical nature of the two antennae, unless specifically stated, the presented data include measurements from only the left JO, while the data from the right JO serve only as a control.

### 2.5. Positive Feedback Stimulation

The essence of the positive feedback stimulation is a feedback loop established using the amplified in-phase response of a sensory unit as the signal to drive the stimulation loudspeaker. Applying such a stimulation to the sensory unit, we expect it to ‘sing’ at the frequency which is close to its intrinsic tuning frequency—an effect hereafter called ‘autoexcitation’. According to our view, several differently tuned PSNs near the electrode tip compete to set the autoexcitation frequency (AF). When one of them (in some cases two or, rarely, three) puts the system into the autoexcitation, other sensory units with different tuning frequencies become inhibited due to the general level adaptation of the JO. Many similarly tuned units oriented in line with the stimulation sound wave may be active during the autoexcitation, but the ability to determine the AF must be extremely sensitive to the amplitude of a unit’s recorded response, which in turn depends on the distance to the recording electrode.

The value of the AF was inevitably affected by the total phase shift in the feedback loop, arising primarily from the latency of the sensory cell response. As the phase shift was not constant along the frequency range, for each frequency measurement, a compensatory phase shift was introduced into the loop. The value of that phase shift was adjusted to provide the maximal amplitude and stability of autoexcitation at the same level of feedback.

The stimulation signal passed through a complex frequency filter to compensate for the influence of the loudspeaker mechanical resonance at lower frequencies. Additionally, the signal was low-pass-filtered with a slope of −6 dB per octave above 500 Hz to reduce the second and higher harmonic components of the sensory unit’s extracellular response. The stimulation signal was limited within an amplitude of 80 dB SPVL to prevent its uncontrolled rise due to positive feedback.

The complete feedback loop included the following elements: the JO sensory unit and its axon in the antennal nerve, recording microelectrode, amplifier, stimulation amplitude adjuster, frequency filter, phase adjuster, amplitude limiter, power amplifier, loudspeakers and, closing the loop, flagellum of a mosquito’s antenna that passed vibrations to the JO.

The method of positive feedback stimulation may be considered a functional sharpening of the electrode, allowing for the recording of only a single or a few axons of the PSNs lying in immediate proximity of the electrode tip. To our knowledge, this method currently allows the best spatial resolution of recording from multiple tightly packed sensory neurons.

Before making any tests of a unit’s directionality, we ensured the quality of recording according to the following criteria: low levels of feedback required to start the autoexcitation and the presence of not more than three different units responding in a narrow angular range.

Positive feedback stimulation allows us to separate the antiphase responses of the axons of sensory units which are, in most cases, recorded together [13,30,32] and are presumed to lie close to each other in the antennal nerve. Contrary to this, other kinds of acoustic stimulation (e.g., sinusoidal stimulation) do not allow such a separation, as each of the units respond to both phases of the acoustic vibration. On the other hand, feedback stimulation does not allow us to measure the true auditory threshold of a sensory unit, providing only a relative estimate of sensitivity. Thus, for each recording site, we used two kinds of acoustic stimulation: positive feedback and sinusoidal.

### 2.6. Measurements of Directionality

The successive stages of the electrophysiological experiment are summarized in Figure 3.

Two kinds of directional measurements were performed, depending on the type of stimulation. (1) The thresholds of the feedback stimulation were measured depending on the direction of a sound wave, producing an unipolar plot (hereafter ‘polar pattern’, example in Figure 4C). (2) Using the measured AF value as the frequency of sinusoidal stimulation, the absolute auditory thresholds were measured depending on the direction of a sound wave, producing a bipolar plot (hereafter ‘directional characteristics’, example in Figure 4D).

In 162 mosquitoes studied, of which in 24 the auditory responses were recorded from the right JO, a total of 290 (50 from the right JO) polar patters were obtained (Table 1).

In the mode of feedback stimulation, we could only measure the relative threshold of auto-excitation for the entire system, including the mosquito and the stimulation setup. Such a relative threshold was defined as the signal level, which required one more incremental step (+1 dB) at the attenuator output for the system to enter a state of sustained autoexcitation (Figure 2D). For each AF, we measured a polar pattern (relative threshold of autoexcitation as a function of orientation of the acoustic wave vector relative to the axis of the antenna, with a 15° step). Then, using sinusoidal stimulation at AF (rounded to 5 or 10 Hz), a directional characteristic (similar function for auditory threshold).

The criterion of the response threshold was set at 2 dB of sustained excess of the response amplitude above the average noise level in a given recording. At each combination of stimulation parameters, the threshold was measured consecutively at least twice. Such auditory thresholds, measured in response to sinusoidal stimulation and expressed in dB RMS SPVL, are hereafter called ‘absolute thresholds’ or just ‘thresholds’.

Similarly, after finding the best direction using the positive feedback stimulation, we measured a frequency-threshold curve in the range of 40−300 Hz using sinusoidal stimulation with a 5 Hz step below 170 Hz and a 10 Hz step otherwise. Below 40 Hz, the measurements were limited by a substantial decrease in the efficiency of the speakers.

The best frequency and best direction of a given unit, obtained by means of measuring the auditory thresholds, also served as a control check for the AF and best direction, obtained through a positive feedback stimulation.

### 2.7. Search for Low-Frequency Units

Based on the preliminary results, we realized that the true responses of the low-frequency sensory units are often difficult to distinguish from the combination harmonics that appear when two or more high-frequency units respond simultaneously at different frequencies (example of two responses in Figure 2D, traces at 69 and 114 Hz). This effect could lead to the under-estimation of true low-frequency-tuned units. Moreover, the positive feedback stimulation method was found to be rather ineffective at the lower frequencies, most probably due to the high competition from the more sensitive units tuned to higher frequencies, which often captured the autoexcitation frequency. Accordingly, we designed an additional series of experiments to compensate for these biases. In these experiments, the searching frequency was set to 70 Hz, and during the choice of the recording site we gave preference to the units that responded maximally at that or at lower frequencies. Moreover, the initial (searching) vector of the acoustic wave was set to φ = –45°, that is, to search for units in the II and IV quadrants, where, according to the preliminary analysis of the data, we expected to find less responding units.

### 2.8. Data Analysis

Directional plots. In an array of threshold data obtained from a single recording site, we determined the maximum threshold value (Th _max_). Based on this, a set of derived values describing each unit’s directional characteristic or polar pattern was estimated using the equation
A_i_ = Th _max_ − Th _i_,(4)
where Th _i_ is the threshold at any given direction of the sound wave. In the plots based on these data, the sectors of the highest sensitivity corresponded to the lowest recorded thresholds, and the central zero point corresponded to Th _max_. The angles at which no response at the best frequency was observed were given the value A_i_ = 0.

The angular sensitivity range of a unit (the width of its directional characteristic) was determined at −6 dB of its maximum sensitivity (in case of bipolar directional characteristics the values from the two independently measured symmetrical plots were averaged). The best direction of a given unit was determined as the bisector of this range.

Sonograms, from which the AF data were extracted, were plotted and analyzed in Sound Forge Pro 10 (Sony).

Correction for temperature. As the experiments were performed in a range of ambient temperatures, all frequency data were normalized to 20 °C according to the previously established equation:F_20_ = F_t_ (1 + k (20 − t)), (5)
where t is a temperature for a given experiment and k = 0.02 is a relative shift of tuning frequency with a change of frequency by 1 °C [34].

Statistics. Statistical analysis was performed in PAST [35] and R [36]. Measurements are given as mean ± standard deviation, with the distributions pre-tested for normality using the Shapiro–Wilk test.

## 3. Results

### 3.1. Position of the Antennae in Flying Mosquitoes

The angle at which a flying mosquito holds its antennae relative to the horizon was 35.5 ± 6.7° (n = 66), with no correlation (Pearson ρ = 0.037) to the angular position of the body, measured along the abdomen (43.3 ± 7.8°; Figure 1A). The variation of these parameters was not significantly different (F test, *p* = 0.21). The projection of the inter-antennal angle to the horizontal plane was 76.8 ± 9.0° (n = 63, Figure 1B), which corresponds to the average value 65.7° in the plane of the antennae, according to Equation (1), Section 2.2.

### 3.2. Individual Directional Characteristics of Sensory Units

A typical threshold directional characteristic of an individual sensory unit measured with the sinusoidal pulse stimulation was symmetrically bi-directional, showing a classic figure-eight pattern (Figure 4D). The width of each petal was 121 ± 18.6° (n = 96).

Acoustic stimulation in the positive feedback mode allowed us, as in previous studies [13], to discriminate several different responses at each recording site. First, the feedback stimulation combined with directionality test revealed unidirectional polar patterns (Figure 4C). With slight deviations (±8°), they overlapped with one of the petals of the bi-directional characteristics measured at the same site using sinusoidal stimulation (Figure 4D). These results were fully expected, as one and the same sensory unit should demonstrate exactly such responses: the phase of the stimulation matters in the positive feedback stimulation but not in the sinusoidal mode, where a unit cannot discriminate the two opposite directions. This should not be confused with the two different units, each possessing individual frequency tuning, responding at the same recording site (see below).

In 54 recordings (23% of the total) there existed only a single unilateral response to a feedback stimulation, not complemented with other responses at different phases (example in Figure 4C).

In most recordings (n = 162, or 67%) there appeared another response with a different AF (example in Figure 4E). Its polar pattern was oriented in opposition (180 ± 10°) to the first one. In such recording sites with an anti-phase pair of units, sinusoidal stimulation revealed similar thresholds of response (ΔTh ≤ 3 dB) to both AFs determined by the feedback stimulation or to the intermediate frequency of stimulation (Figure 4F).

In rare cases, there was simultaneous auto-excitation at two different frequencies within a single polar pattern and at a third frequency in the opposite direction (n = 6) or without any response in the opposite angular range (n = 3).

### 3.3. Radial Distribution of Sensory Units in the JO

Figure 5A shows the distribution of the best directions measured in the sensory units of the left JO. Figure 5C shows the same data plotted in polar coordinates centered at the axis of the left antenna. Recorded auditory units are distributed unevenly: the number of units oriented in the first and third quadrants is 3.2 times higher compared to the second and fourth quadrants (181/56, χ^2^ = 33, *p* < 0.001). The same bias persisted in the subset of the data when the mosquito was initially rotated by 90° relative to the stimulation system and to the recording electrode holder (Figure 5A, black-filled part of the distribution, angular data corrected, respectively). The control series of measurements made from the right JO revealed the majority of units were oriented in the second and forth quadrants (31/9, χ^2^ = 6.05, *p* < 0.025, Figure 5B), which mirrors the distribution obtained from the left JO.

In the distributions of best directions of units in both left and right JOs (Figure 5A,B) there were two additional sub-peaks at 80–90° and 260–270°. These units, oriented in the antennal plane (see Figure 1A,B), constituted 17% (n = 42) of the total number of recorded units. Their proportion was significantly higher than that of the units oriented perpendicular to the antennal plane, 0 ± 10° and 180 ± 10° (n = 15; χ^2^ = 6.4, *p* < 0.025).

Sinusoidal stimulation allowed us to measure the absolute thresholds of the units, which are plotted in polar coordinates in Figure 5C. Due to the symmetrical nature of the directional characteristics, each data point on the plot had its pair located symmetrically relative to the center of coordinates. The individual thresholds of the recorded units varied from 30 to 53 dB SPVL, with 39.8 ± 4.8 dB SPVL on average (n = 96); their distribution is shown in Figure 6E. Most sensitive units with thresholds of 36 dB SPVL or lower were found around 53° (Figure 5C), with tuning frequencies in the range of 65–135 Hz.

The data from a separate experimental series (70 Hz, –45° searching stimulation, which gave preference to lower frequency units oriented in second and fourth quadrants), despite the searching preference, also demonstrated that most recorded units were oriented in the first and third quadrants (27/6, χ^2^ = 6.68, *p* < 0.01, n = 25, Figure 5D), i.e., similarly to the main array of data for the left JO (Figure 5C).

Polar distribution of AFs did not show significant asymmetry in the frequency tuning around the JO axis (Kruskal–Wallis test for distributions of AFs in four quadrants, *p* = 0.82; Figure 5E,G). On the frequency axis, the total distribution of AFs (Figure 5F) is neither normal nor log-normal (Shapiro–Wilk test). It contains two local components that differ from the theoretical curve: minimum at 10**^1.94^** = 87 Hz (χ^2^ = 4.03; *p* < 0.05) and maximum at 10**^2.1^** = 126 Hz (χ^2^ = 11.24; *p* < 0.001).

### 3.4. Audiograms

Most of the audiograms measured from either the left or right JO can be divided into three groups: (i) with a slope towards the lower frequencies, (Figure 6A); (ii) with the main minimum at 50–60 Hz (Figure 6B); and (iii) with the main minimum at 90–100 Hz or higher, which was observed in most cases (Figure 6C). The division between the latter two groups is rather arbitrary, as there was a continuous variety of frequency tuning, most probably arising from the summation of responses from a number of differently tuned units in the compound extracellular recording. Such summation is rarely allowed to measure the Q-factor of a single unit. However, we managed to do it in several recordings of higher quality: examples of audiograms measured in such experiments are shown in Figure 6D together with the Q-factor values estimated at +6 dB from the minimum threshold.

### 3.5. Ratios between Individual Frequencies in Pairs and Triplets of Units

Although this did not relate to the main scope of this study, after obtaining a significant dataset (n = 240) of the individual frequencies (AFs) of the JO auditory units, most of which were measured pairwise within the same recording sites, we analyzed the frequency ratio in these pairs. A regular non-random nature of the ratio would speak in favor of interactions between the paired units, also confirming our previous findings in male mosquitoes [13].

The histogram of the ratio between AFs in pairs of anti-phase units F_2_/F_1_ (F_2_ > F_1_) has a distinctly regular structure (Figure 6F, n = 81). The distribution is asymmetric as it is limited to below 1. Taking the logarithm of the data removes this restriction (Figure 6G). The sum of the main components Σ χ**^2^** corresponds to the four major peaks of the original distribution: F**_2_**/F**_1_** = 10**^0.096^** = 1.25 (χ**^2^** = 8.77; *p* < 0.01), F**_2_**/F**_1_** = 10**^0.125^** = 1.33 (χ^2^ = 30; *p* < 0.001), F**_2_**/F**_1_** = 10 **^0.175^** = 1.5 (χ^2^ = 31; *p* < 0.001), and F**_2_**/F**_1_** = 10**^0.225^** = 1.67 (χ**^2^** = 28; *p* < 0.001). The values of the peaks are either equal or close to the integer fractions 5/4, 4/3, 2/3, and 5/3. We did not have enough data for triplets of the units (n = 6) to perform similar analysis, however, the ratios between individual frequencies demonstrated the same pattern.

### 3.6. Directionality of Sensory Units in Three Dimensions

In a three-dimensional space, the transition from a single sensory unit to the system of many units belonging to the left and right antennae and forming the common auditory space of a mosquito may be difficult to imagine. To simplify the task, below we present this transition in sequential steps. Additionally, in the Appendix A we provide a video file and a parametric computer model that graphically illustrate the steps described below.

Positive feedback stimulation enables us to separate the activity of paired antiphase units. Under sinusoidal stimulation, both units possess similar sensitivity to acoustic signals coming from the opposite directions (Figure 4C–F). The resulting three-dimensional audiogram of a single JO sensory unit is a combination of the mechanical directional diagram of the antenna and the directional characteristic of a unit in a plane perpendicular to the antenna. The diagram of the antenna in the first approximation has the shape of a torus with minima on the apex and the base and the maximum in the plane perpendicular to the flagellum (Figure 7A). An averaged two-dimensional directional characteristic of a unit consists of two symmetrical petals of more or less circular shape (Figure 4D,F), shown in dark-grey in Figure 7A. Then, the simplified three-dimensional characteristic of a single sensory unit would be a system of two contacting spheroids with their common axis oriented at an angle (*φ)* to the transverse axis of the JO.

For the sake of initial simplicity, let us temporarily assume that the paired antennae are held horizontally and parallel to each other. Then, characteristics of the two symmetrically oriented units, belonging to the sectors that are populated with the most sensitive units, φ = 53° on average (φ= –53° for the right JO), will overlap as shown in Figure 7B, with the mosquito viewed from the front; characteristics L1+L2 and R1+R2 belong to the units of the left and right antennae, respectively.

The antennae of a female mosquito in their average in-flight position are spread apart to an angle, 2ξ (Figure 7С). By spreading the antennae, the parts of the directional characteristics L1 and R1 shift forward and medially, while L2 and R2 shift backward and medially (Figure 7D,F). With such a shift, the overlap between the left and the right characteristics increases proportionally to the angle 2ξ (Figure 7D,E). It may seem counter-intuitive, but the wider spreading of the antennae increases rather than decreases the overlap between the auditory spaces of the left and right auditory units, at least of the most sensitive units oriented at φ = –53° (right JO) and φ = 53° (left JO). Higher overlap may enhance binaural hearing based on the so-called equisignal zones (see below). However, the spreading of the antennae leads to the skew of the auditory space in the lateral view (Figure 7F).

The upward inclination of the antennal plane relative to the horizon (angle β, Figure 7C) can compensate for the skew of the auditory space at the previous step: the dorsal parts of the directional characteristics (L1 and R1) shift backward, while the ventral parts (L2 and R2) shift forward (Figure 7G). Geometrically, the condition of full compensation, when the dorsal auditory space is located strictly above the ventral one, is described by the following equation:β = arctg (tg φ · sin ξ).(6)

By substituting the measured inter-antennal angle in flying mosquitoes into this equation, 2ξ = 65.7° (ξ = 32.8°), and the orientation of the most sensitive units, φ = 53°, we get the angle β equal to 35.7°. This is very close to the measured value of β, 35.5 ± 6.7°.

### 3.7. Equisignal Zones

The most remarkable feature of the overlap between the left and the right directional characteristics is that they form the so-called equisignal zones (shown in dark grey in Figure 7D,E). Within these zones, the angular position of a sound source relative to a mosquito can be estimated from the ratio between the amplitudes of the responses coming from the left (A_left_) and the right (A_right_) units, where (A_left_ − A_right_) represents IAD:M = (A_left_ − A_right_)/(A_left_ + A_right_).(7)

The functional advantage of the parameter M is that, within the equisignal zone, it does not depend on the absolute level of sound. Remarkably, the overlap of directional characteristics of the left and right antennae, at least those of the units oriented at φ = 53°, become higher when antennae are spread apart than when they are brought together (Figure 7D,E). However, such a simplified scheme does not allow us to unambiguously determine the direction to the sound source: the sounds coming from each of the four equisignal zones (Figure 7B) can result in similar IADs.

## 4. Discussion

### 4.1. Functional Asymmetry of the JO

All the evidence we have to date indicates that the auditory function of the mosquito JO is asymmetrical: although its morphology demonstrated radial symmetry [10,37], most of the units responding to sound were found in two of four quadrants (I and III of the left JO, Figure 5). Recording from the contralateral (right) JO revealed a similar but mirrored distribution (Figure 5B), contesting the hypothesis that the observed asymmetry was an artefact. At the same time, each JO possessed units oriented in two other quadrants (II and IV) which, given the three-dimensional geometry of the two antennae, may duplicate the function of the contralateral units.

After finding that quadrants II and IV of the left JO are depleted in sound-responding units (Figure 5C,E) we tested a hypothesis that would explain such functional asymmetry in a morphologically symmetrical system. We assumed that the II and IV quadrants of the JO are populated primarily by low-frequency units tuned below 80 Hz, which we could skip due to the inadequate frequency of the searching signal (100–130 Hz). The presence of sensory neurons tuned to low frequencies in *Drosophila* along with the neurons tuned to higher frequencies [38] spoke in favor of this hypothesis.

However, additional experimental series with a low-frequency searching signal (70 Hz) partially discarded this hypothesis: the angular distribution of low-frequency units (Figure 5D) did not show significant differences from that of high-frequency units (Figure 5C). It should be noted that, in this study, the low-frequency range of stimulation extended down to 40 Hz only, while in *Drosophila*, low-frequency-tuned neurons were detected at 19 Hz or even lower due to a different method of stimulation. Many frequency-threshold curves that we obtained suggested that some ofthe neurons were tuned to frequencies lower than 40 Hz (Figure 6A).

Assuming that the mosquito JO is morphologically symmetrical, it is still unclear what was the functional role of the remaining sensory units, which belonged to the quadrants II and IV and did not respond to our stimulation. Studies in *Drosophila* suggest that not all JO neurons are used for hearing: only about half of the JO neurons responded to antennal vibrations and mediated hearing, while the other half detected the antennal deflections imposed by gravity and wind [38,39,40].

### 4.2. Optimal Position of the Antennae in Flying Mosquitoes

The lack of correlation between the variable in-flight position of a mosquito body and the angle at which it holds the antennae relative to the horizon (Figure 1A) suggests that there is a mechanism that allows a mosquito to control the in-flight position of the head and/or antennae. This finding is trivial, as a flying animal normally relies on world-centric rather than on body-centric coordinates. The head movements to stabilize gaze were described in tethered flies [41,42] and are almost certainly present in mosquitoes. Within the scope of this study, it is only important that the comparatively low variation of the antennae position relative to the horizon allows for world-centric directional hearing.

Since the best direction (φ) is very different among the JO auditory units, there is no position of the antennae that would be optimal for all units in order to provide binaural hearing. However, we may assume that an increased concentration of auditory units or their higher sensitivity at certain sectors of the JO reflects the areas of special interest of a mosquito. Then, the angular distribution of units (Figure 5C,D,E) together with the most frequently observed orientation of antennae (Figure 1) suggests that female mosquitoes pay special attention to the sounds coming from above and from below (Figure 7G). The upward inclination of the antennae is well coordinated with the inter-antennal angle and the distribution of the most sensitive auditory units in the JO.

The second largest group of units was oriented in the frontal plane (φ ≈ 90° and φ ≈ 270°, see sub-peaks in the distribution, Figure 5A). When the antennae are spread apart, parts of the directional characteristics of these units are shifted forward, forming another equisignal zone directly in front of the mosquito.

However, as the position of the antennae may vary in different behavioral states, a mosquito can constantly adjust the parameters of its directional hearing depending on the current sensory task.

### 4.3. Uncertainty in the Sound Localization

As already stated, due to the symmetry of a single unit’s directional characteristic relative to the antenna flagellum (Figure 4D), there exists an uncertainty in the localization of the sound source in the two opposite directions. Earlier, Belton [7] pointed out that the phase of oscillations in both antennae may be significant for directionality. A point source of sound at an angle greater than 40° from the antennal plane would deflect both antennae in phase, while a sound source directly in front of the antennae would deflect the two flagella out of phase. It is quite possible that the mosquito brain performs a combined analysis of the amplitude and phase ratios of the signals coming from the right and left JOs to reduce uncertainty in estimating the direction to the sound source.

### 4.4. Frequency Tuning

According to the combined distribution of tuning frequencies (Figure 5F), the majority of audiograms (Figure 6A–C), and the number of recorded narrow-band units (Figure 6D), the hearing of *Culex pipiens* female mosquitoes ranges up to 260 Hz. Low-frequency units (40–80 Hz) were found to be 6 dB less sensitive on average, but their minimal thresholds (ca. 30 dB SPVL) were similar to the units tuned above 80 Hz. Thus, the ability of female mosquitoes to hear low frequencies extends down to 40 Hz (and, most probably, even lower) without a significant decline in sensitivity.

Earlier it was shown that the JO of *Culex* female mosquitoes possesses low-frequency sensory units that significantly decrease their thresholds (by 26 dB on average at 40 Hz) when affected by the signal simulating the flight vibration, i.e., when a second dorso-ventral acoustic stimulus was applied as a background [20]. It is possible that female mosquitoes possess a specialized set of low-frequency-tuned PSNs that are intended for the in-flight reception of some specific signals, such as vocalizations or sounds accompanying the movements of host animals.

Similar to male mosquitoes [33], the female JO contained a set of narrowband (with Q_6_ about 6) units distributed along the whole frequency range of their auditory sensitivity (Figure 6D). The thresholds of these units were higher than those of the majority of recorded broadband units (compare to Figure 6A–C). Due to this difference in sensitivity, narrowband units were only rarely recorded, as their responses were in most cases masked by the activity of more sensitive broadband units.

### 4.5. Interactions between the Paired Auditory Units

The analysis of the individual frequency tuning of the JO auditory units was beyond the main scope of this study, however, in the course of directional measurements we obtained a large dataset of AFs of the JO neurons, which we report here.

The observed clearly discrete nature of the distribution of frequency ratios in paired systems of units (Figure 6F,G) indicates an electrical or mechanical interaction between the units in a pair. This phenomenon can be explained by the fact that the auditory receptors of the JO in mosquitoes possess the mechanisms of endogenous activity, or generators, that help to amplify low-amplitude signals [43,44].

When two generators, tuned to different frequencies, become coupled, this leads, with a high probability, to a phenomenon called harmonic synchronization. Such an interaction manifests itself in the mutual adjustment of the frequencies of both generators in order to reach some integer ratio between the frequencies. This state corresponds to a local energetic minimum of the system and contributes to stability of the harmonic synchronization [45]. Thus, the observed mutual synchronization of auditory units in the mosquito auditory system may occur due to a mechanism which is common in nature and is not necessarily a sign of specific adaptation. Affected by some destabilizing factors, a pair of coupled generators can switch to different modes of synchronization, for example, changing a frequency ratio from 3/2 to 4/3. In addition, the range of possible frequencies which a generator can be tuned to can have biophysical constraints, in that way additionally limiting the choice of a frequency ratio. Most probably, the combination of all these factors leads to the appearance of a set of integer frequency ratios in the paired systems of mosquito auditory units: 5/4, 4/3, 3/2, and 5/3 (Figure 6F). Morphologically, sensillae that include only a single receptor cell have not yet been described in the mosquito JO; all auditory sensillae contain at least two mechanoreceptor cells [10]. Meanwhile, a significant probability of recording single units (23%) implies their wide representation in the JO. It can be assumed that the activity of such ‘single units’ is a synchronous response of two units tuned to the same frequency (F_2_/F_1_ = 1) and belonging to a single sensilla.

The functional significance of paired auditory units may have different but complementary explanations. The system of two in-phase paired units may have an improved signal-to-noise ratio, as the total noise of the system is summed up as a root–mean–square, while the in-phase signals are linearly summed, thus increasing the signal-to-noise ratio of the paired system by 2/√2 = √2, or 3 dB in terms of a threshold. We presume that, in an anti-phase pair, all oscillations beyond the frequency ranges of the two units, including the low-frequency deflections caused by wind currents, should be mutually subtracted. Thus, the anti-phase pairs of the units, which are abundant in the mosquito JO, may represent a system of selective frequency filters.

### 4.6. On the Possible Functions of Hearing in Female Mosquitoes

This is another study that demonstrates a very developed auditory system in female mosquitoes. Even if female hearing had initially evolved to enable acoustic interactions with their male counterparts, there are reasons to believe that blood-sucking mosquitoes do use acoustic cues for host-detection, including the sounds of human speech. At the very least, nothing would prevent a female mosquito from hearing the sounds in the low-frequency range (up to 260 Hz for *Culex pipiens*) and to estimate and follow the direction to its source. There is indirect evidence in favor of the host-detecting function of hearing in female culicids: in a different family of dipterans, Chironomidae, which has also evolved extremely sensitive hearing in swarming males [46,47,48] but no blood-sucking in females, any data on the female hearing are lacking. 

## 5. Conclusions

According to our findings, female mosquitoes possess asymmetric distribution of auditory neurons within the Johnston’s organ, mirrored in the left and right. The neuronal asymmetry is coordinated with the in-flight position of antennae to maintain a uniformly shaped auditory space, symmetric relative to a mosquito body. This finding may reflect a common principle of directional hearing based on the particle velocity receivers, which in its turn may determine the ranges of angular position of antennae in different insects.

Within the mosquito auditory space, there are areas located above, below, and in front of a flying mosquito, which we called equisignal zones. Within these zones, directional cues should not depend on the absolute level of sound, and, accordingly, on the distance to the source. This feature may be critically important during swarming behavior, when the everchanging distance to other mosquitoes demands instant and precise directional estimates.

At the same time, each JO possesses units that duplicate the function of the contralateral one. Such redundancy increases the reliability of the entire system and potentially can provide acoustic orientation even if one antenna gets damaged.

The auditory system of female mosquitoes has enough resolution to estimate the direction to the sound source, while its frequency range enables detection of sounds produced by the animal and human hosts.

## Figures and Tables

**Figure 1 insects-14-00743-f001:**
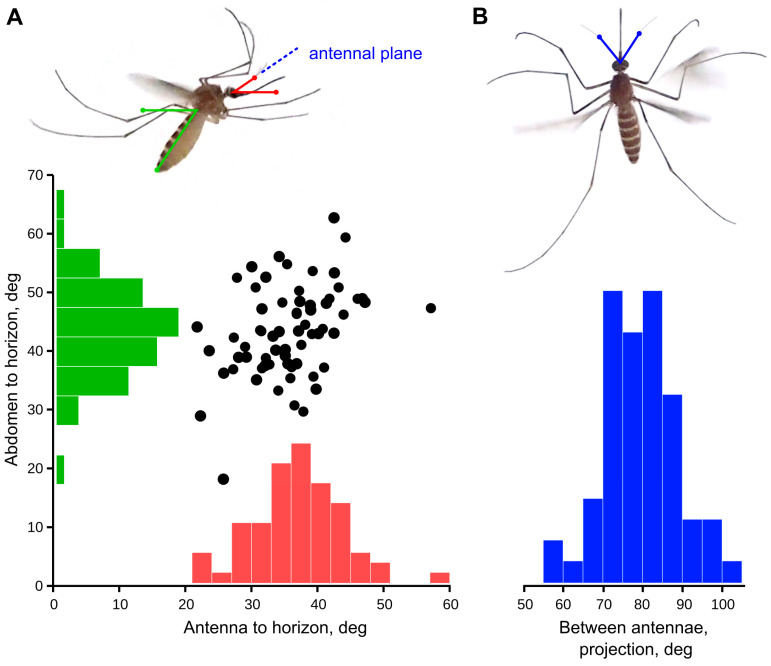
Orientation of antennae in a flying mosquito. The measurements were taken from multiple still images of flying mosquitoes, photographed in lateral (**A**) and dorsal (**B**) aspects. There was no clear dependence between the angles of antenna-to-horizon (red) from the abdomen-to-horizon (green), suggesting that the orientation of antenna to the world rather than to the body matters to the mosquito. The true inter-antennal angle (2ξ) was calculated from the measurements in horizontal projection (α, blue) according to the Equation (1) (see Methods).

**Figure 2 insects-14-00743-f002:**
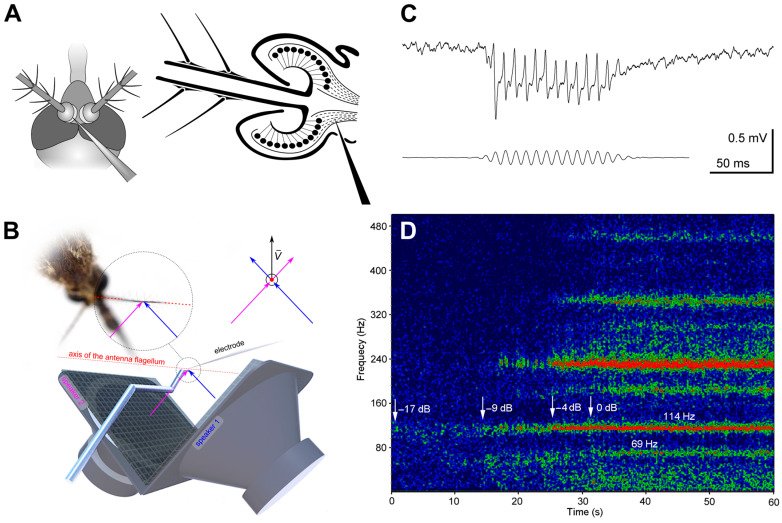
Method of measurement of the JO directionality in mosquitoes. (**A**) Schematic view of the insertion of recording glass microelectrode through the scape–pedicel joint into the antennal nerve. (**B**) Vector superposition of the signals from two orthogonally oriented speakers at the point of the antenna. The mosquito is positioned at the crossing of the axes of two speakers. The axis of the antenna′s flagellum (red dotted line) is perpendicular to the directions of the sound waves (magenta and blue arrows) produced by both speakers. On the right, the same schematic is shown as viewed from the base of the antenna along its flagellum. The resulting direction of the air vibration velocity (v) that affects the antenna is determined by the vector superposition of two signals, allowing us to set the desired direction for the acoustic stimulation. (**C**) Extracellular response to sinusoidal stimulation recorded from the axons of the antennal nerve. Voltage scale is given for neuronal response. Lower trace: stimulation signal. (120 Hz, 58 dB SPVL), recorded by the microphone located 2 cm from the mosquito. (**D**) Sonogram (frequency spectrum of signal against time) of response to a positive feedback stimulation, when the electrical potential recorded from the antennal nerve is amplified and fed to the speaker in real time. Colour represents the relative amplitude of response, from blue to red. The recording starts in silence (−17 dB below the threshold), then the level of feedback is gradually increased, indicated by arrows with respective values of the feedback level. The traces of the feedback effects appear at −9 dB (narrow-band frequency-selective increase in noise). At 0 dB (threshold) there appears a sustained autoexcitation at 114 Hz and a weaker autoexcitation at 69 Hz. Two traces of simultaneous excitation at different frequencies (69 and 114 Hz) represent differently tuned sensory units. Traces at higher frequencies are harmonics.

**Figure 3 insects-14-00743-f003:**
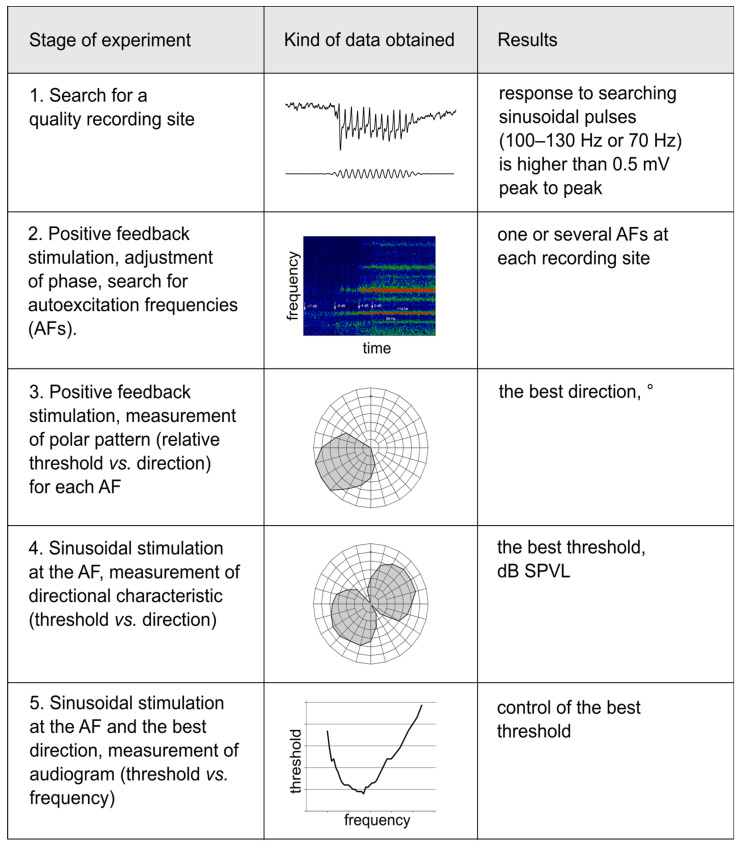
Measurements of the directionality of the JO auditory units, design of experiment. The sequence of measurement stages performed for each auditory unit is shown in the left column. The central column shows examples of data obtained at each stage. The right column lists the main results obtained at each stage. Stage 1: Methods, 2.3; example in Figure 2C. Stage 2: Methods, 2.5; example in Figure 2D, Figure 5E−G, Figure 6F,G. Stage 3: Methods, 2.6; example in Figure 4C,E, Figure 5A,B. Stage 4: Methods, 2.6; examples in Figure 4D,F, Figure 5C,D, Figure 6E. Stage 5: Methods, 2.6; Figure 6A−D.

**Figure 4 insects-14-00743-f004:**
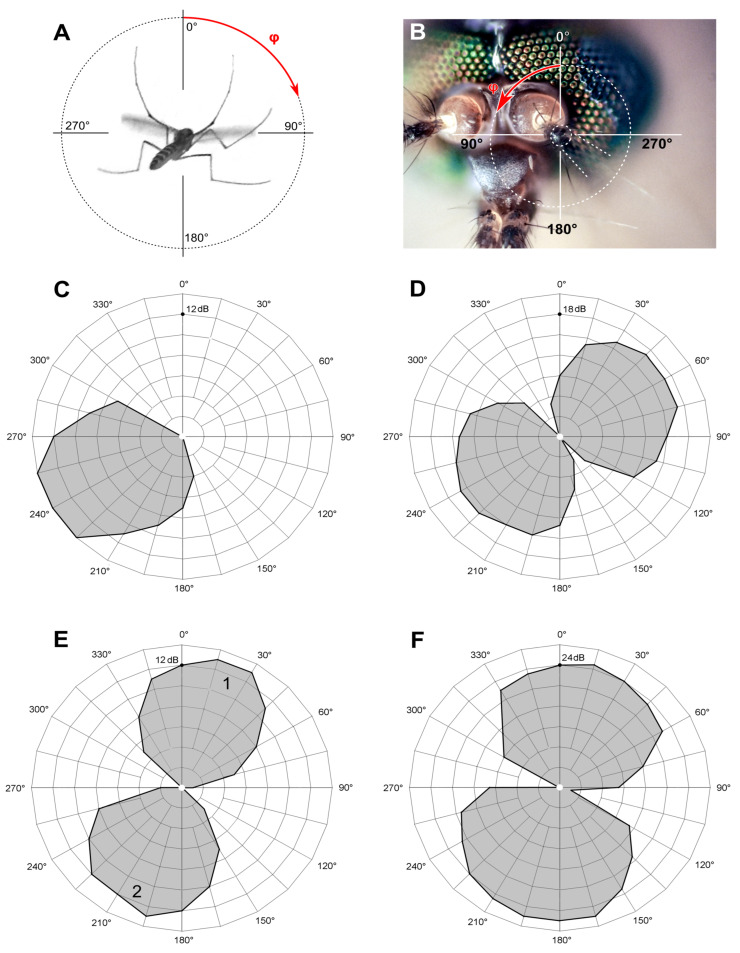
Directional properties of the JO auditory units. (**A**) The polar coordinate system is centered at the left antenna, and the mosquito is viewed posteriorly. The left antenna is hidden behind a mosquito body and is oriented perpendicular to the image plane. The red arrow shows the positive (clockwise) direction of the angular axis used in all polar diagrams for both left and right JOs. (**B**) Photo of a mosquito head shows the same coordinate system, centered at the left antenna. (**C**) Polar pattern of a single responding unit (AF 112 Hz), consisting of one unipolar petal. (**D**) Directional characteristic of the same unit measured by sinusoidal stimulation, containing two almost symmetrical petals (measured independently); threshold of response 32 dB SPVL at 112 Hz. (**E**) Polar patterns of two units recorded at the same site and responding in anti-phase; the tuning frequencies are 104 Hz (unit 1) and 77 Hz (unit 2). (**F**) Directional characteristic of the same pair of units as in (**E**), measured as a threshold to 100 Hz sinusoidal stimulation. (**C**,**E**) Relative sensitivity (inverse of relative threshold) is plotted radially in 2 dB steps, connected by a thick black line. (**D**,**F**) Absolute sensitivity (inverse of absolute threshold) is plotted radially in 3 (**D**) and 4 (**F**) dB SPVL steps, connected by a thick black line.

**Figure 5 insects-14-00743-f005:**
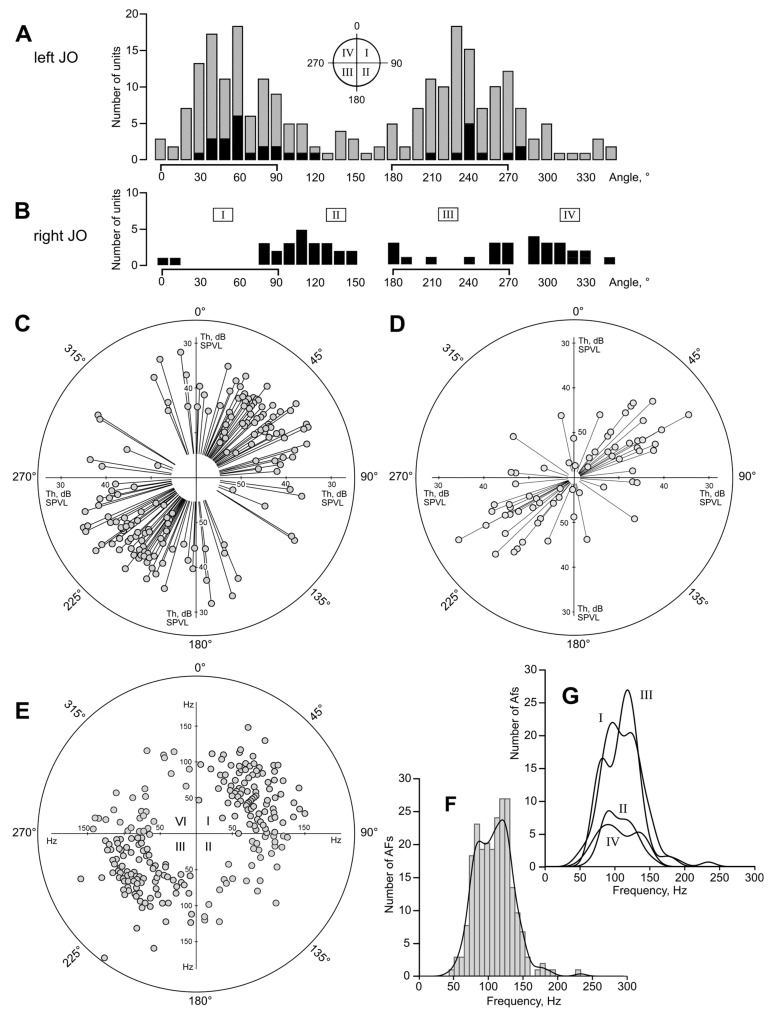
Distributions of auditory units around the axis of the JO. (**A**) Left JO, angular scale same as in Figure 4. Gray bars show overall distribution, black bars represent the data from experiments where a mosquito was positioned ±90° around its antero-posterior axis (the angular data in the measurements were corrected accordingly). Note that, regardless of mosquito orientation and the position of inserted electrode relative to the head and the JO, the majority of sensory units belong to the I and III quadrants (0–90 and 180–270 degrees). The scheme of quadrants is shown above the distribution. (**B**) Similar data obtained from the right JO. Note that the distribution mirrors for the left JO, i.e., the right auditory units are oriented in the II and IV quadrants. (**C**) Polar distribution of individual thresholds, left JO. Lower thresholds (higher sensitivity) are further from the center. Each pair of filled circles follows from a bipolar directional characteristic, as shown in Figure 4D. (**D**) Polar distribution of individual thresholds of low-frequency-tuned units (below 80 Hz), left JO. Note that these units are oriented similarly to the higher-frequency units (**C**), although the initial orientation of the searching stimulation signal favoured the finding of units oriented in the two alternate quadrants (II and IV). (**E**) Polar distribution of autoexcitation frequencies (AFs), left JO. Each filled circle follows from a polar pattern, as shown in Figure 4C. In 11 cases the circles fully overlapped each other. (**F**) Distribution of AFs, n = 240, left JO, with the kernel function shown over the histogram. (**G**) The same dataset as in F, divided into quadrants I–IV, kernel functions of distributions.

**Figure 6 insects-14-00743-f006:**
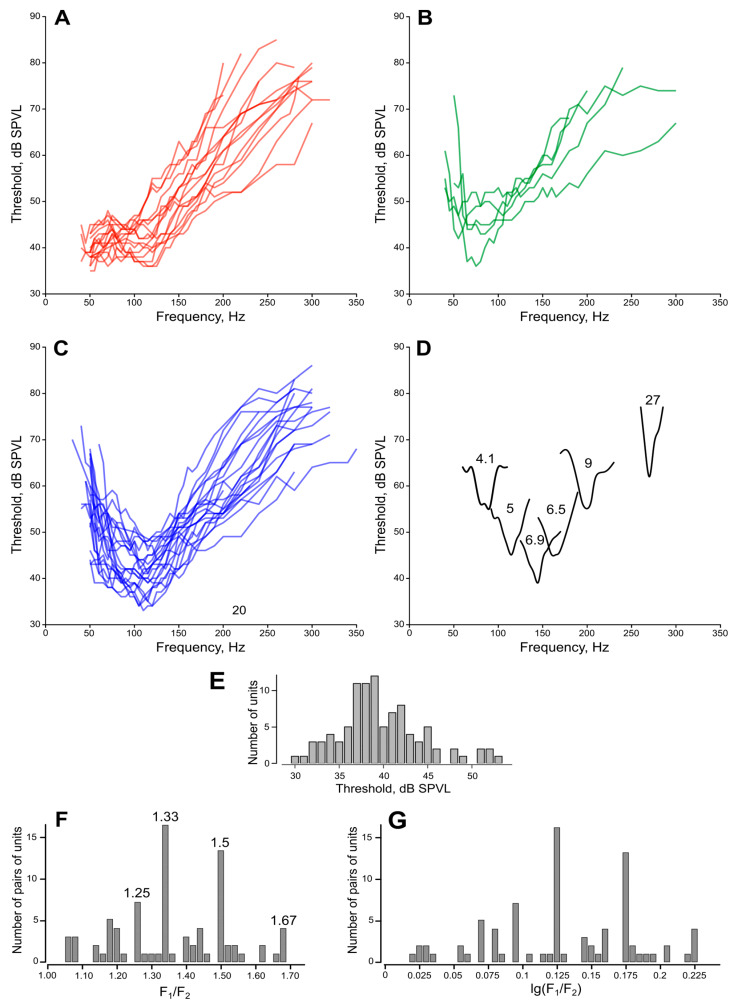
Frequency tuning of the JO auditory units. (**A**–**C**) audiograms (frequency-threshold curves), measured using sinusoidal stimulation. (**A**) slope towards the lower frequencies, the frequency range below 40 Hz was limited by the stimulation equipment. (**B**) Best thresholds at 50–60 Hz. (**C**) Best thresholds at 90–100 Hz or higher. (**D**) Individual curves of narrow-band units measured with a 5 Hz step. Numbers show quality values (Q_6_, measured at +6 dB from the best threshold). Note that such narrow-band units cover almost the whole frequency range of female mosquitoes (40 to 260 Hz) and that their thresholds are mostly not among the lowest ones (compare to Figure 6C). (**E**) Distribution of auditory thresholds of sensory units, each measured at the best frequency and the best direction for a given unit. (**F**,**G**) Ratios of tuning frequencies in paired auditory units, measured by the feedback stimulation. (**F**) Numbers show the values of the most prominent peaks, which correspond to the integer ratios (3/2, 4/3, 5/3, 5/4). (**G**) The same data as in (**F**), logarithmic scale.

**Figure 7 insects-14-00743-f007:**
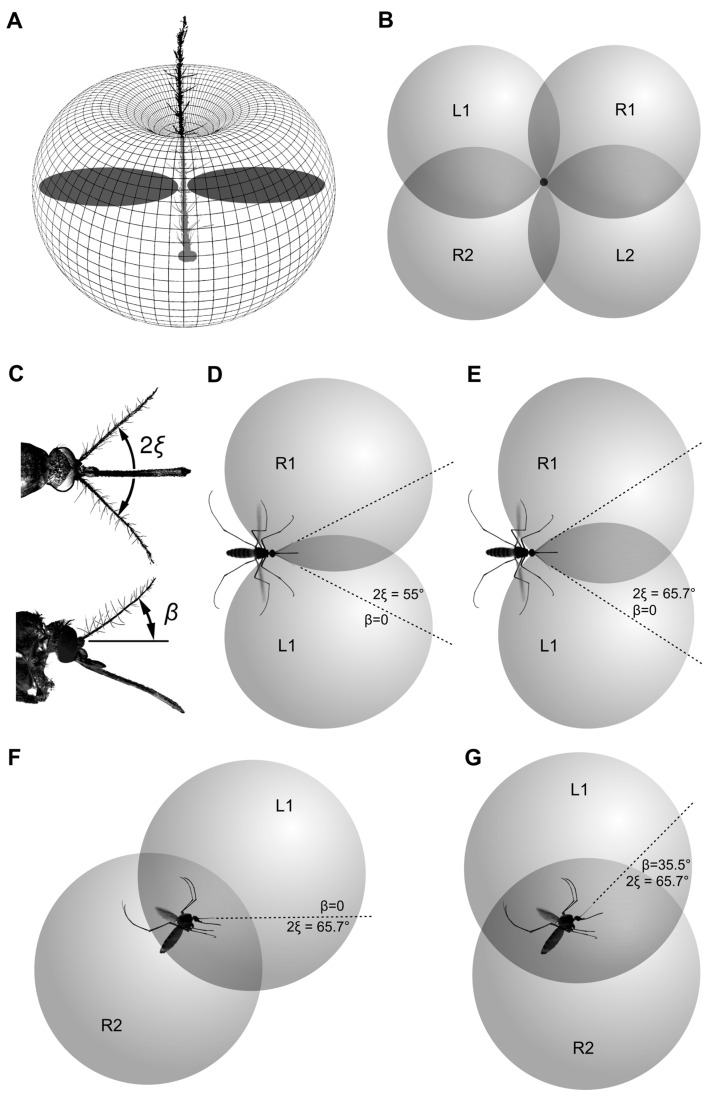
Schematic illustration of the three-dimensional directional characteristics of the two symmetrical groups of sensory units from the left and right JOs, oriented at φ = 53°. (**A**) Theoretical directional diagram of the mosquito antenna. It has the shape of a torus with minima on the apex and the base of antenna and a maximum in the plane perpendicular to the antenna. The flat directional characteristic of an individual unit, as measured in this study, is shown in dark grey. (**B**) Characteristics of left (L1, L2) and right (R1, R2) sensory units when antennae are parallel and oriented towards the viewer (this is non-existent simplification provided for a step-by-step explanation). (**C**) Actual position of antennae in a flying mosquito (see also Figure 1). Antennae are deflected from the parallel in the frontal (angle 2ξ) and sagittal (angle β) planes. (**D**,**E**) Intersection of left and right characteristics when antennae are spread apart (but not raised, β = 0), at different values of an angle between antennae (2ξ) dorsal view. Parts of characteristics L2 and R2 are not shown for simplicity. (**F**) The same, 2ξ = 65.7°, β = 0, lateral view. (**G**) Resulting position of directional characteristics with orientation of antennae as shown in C (2ξ = 65.7°, β = 35.5°), lateral view. Dark-grey intersection areas in (**B**,**D**,**E**) show equisignal zones.

**Table 1 insects-14-00743-t001:** Summary of the electrophysiological directionality tests.

	Left JO	Left JO, Low-Frequency Search (70 Hz)	Right JO	Total
number of mosquitoes/experiments	112	26	24	162
number of individual polar patterns	240	–	50	290
number of individual directional characteristics	96	35	13	144

## Data Availability

The data presented in this study are available in Appendix A and on request from the corresponding author.

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
