# Peer review of "Mapping the Auditory Space of Culex pipiens Female Mosquitoes in 3D"

_insects, 2023, doi:10.3390/insects14090743_

Round 1

Reviewer 1 Report

“Mapping the auditory space of Culex pipiens female mosquitoes in 3D” by Dmitry Lapshin and Dmitry Vorontsov is concerned with the measurement of the orientation of antennae in female Culex mosquitoes during flight and obtaining a detailed physiological map of the directionality of sensory units in the paired Johnston's organs to create a three-dimensional model of the mosquito's auditory space, which the authors discovered was optimal for binaural hearing focused primarily in front of, above and below the mosquito when in flight. The reviewer found the description and rational of the methods to be very comprehensive, the outcome of the experiments to be well presented and the discussion to be justified. The text was hard to follow in places due largely to not defining abbreviations.

Introduction

General points.

This section might be best set up by saying briefly and early on, what it is that female mosquitoes listen to and perhaps the frequency ranges of these sounds. This might make it more understandable why the frequency range of the JO units that are directionally sensitive is quite low. The fact that due to the JO being a particle velocity sensor, and that directional hearing should be at close range, does not appear to be clearly introduced. The problems of directional hearing for small animals are clearly introduced.

Lines 86-91

I think this needs to be more explicit. Is it that the small size of the antennae of female mosquitoes means that they do not obscure the recording site thereby enabling more accurate and repeatable placement of the recording electrode?

Methods

Line 184. Please define SVPL

Line 193. Please define PSNs

Lines 212-212. Does this section refer to a figure? Its rational is hard to follow otherwise.

Lines 217-219 This is hard to understand. It might be more appropriate at the very end of this section (say line 224) because it then makes more sense.

Results

Table 1. Define AFs.

Figure 3 caption. Relative sensitivity is plotted radially in ? dB steps

Figure 4 caption. It would be helpful to have vertical scale bars for A, B, F and G

Line 410 “Ratios between individual frequencies in pairs and triplets of units”. The significance of these measurements is not evident. Please provide a brief rational for making these measurements.

Figure 5 caption. Vertical scales in Fig 5E-G would be helpful.

Lines 435-426. Land discovered this much earlier. M. F. Land, Head movement of flies during visually guided flight. Nature 243, 299–300 (1973).

Line 458. By “ones” do you mean JO sensory units?

The English is good but could be improved in later versions

Author Response

Thank you for clear and helpful review of our paper. Our responses are copied below, however, they are more conveniently presented in the attached pdf file. The Reviewer's comments formatted in Italic.

Introduction

General points.

This section might be best set up by saying briefly and early on, what it is that female mosquitoes listen to and perhaps the frequency ranges of these sounds. This might make it more understandable why the frequency range of the JO units that are directionally sensitive is quite low. The fact that due to the JO being a particle velocity sensor, and that directional hearing should be at close range, does not appear to be clearly introduced. The problems of directional hearing for small animals are clearly introduced.

What it is that female mosquitoes listen to – the honest answer would be "we do not know". We tried to mention all that is known in Introduction: precopulatory interaction with males (rather questionable), attraction to communication sounds of hosts (e.g. frogs), possible attraction to the sounds of human speech. We believe that they use hearing for both tasks (intraspecific communication and host-detection), while the distance range of their hearing may be not so close, as was shown by Menda et al (2019).

The revised version of the Introduction about the function of hearing in female mosquitoes:

The question 'What do female mosquitoes hear?' demands much attention by itself. The auditory system of biting female mosquitoes is poorly studied compared to that of the conspecific males, the outstanding listeners. Since mosquitoes spread dangerous diseases like West Nile fever, the understanding of their hearing system is of crucial importance. Our very limited knowledge on such a practically important subject can be at least partially explained by the fact that for a long time it was difficult to demonstrate any behavioural auditory responses in female mosquitoes, although they also possess a sophisticated auditory system, only surpassed by their male counterparts.

Pre-copulatory acoustic interaction, demonstrated for a number of mosquito species [14–18] suggests that a female mosquito hears the flight tone of a male. Such hearing is possible if a female detect a difference tone produced by nonlinear mixing of its own flight tone with the one of a male [15, 19–22].

Among bloodsucking dipterans, there are examples of distant attraction to the communication sounds of frogs in midges [23–25] and mosquitoes [26–28]. A study by Menda et al. [29] suggests that the Aedes female mosquitoes can be attracted to the sound frequencies similar to those of human speech. Although the results on the female mosquito audition are far from conclusive, from the existing behavioural studies one can reason that bloodsucking females of some dipterans, and mosquitoes in particular, can perceive the direction towards the sound source.

I think this needs to be more explicit. Is it that the small size of the antennae of female mosquitoes means that they do not obscure the recording site thereby enabling more accurate and repeatable placement of the recording electrode?

Thank you, corrected:

Such bias is hard to avoid in experiments on male mosquitoes with their large extended antennae, which limit the direction of an electrode insertion.

The limitation can be overcome by recording from the JO of a female mosquito. The antennal fibrillae of female mosquitoes are smaller and allow to insert the recording electrode from multiple directions, which would significantly decrease the possible bias caused by blind selection of recording sites within the antennal nerve.

Line 184. Please define SVPL

Done

Line 193. Please define PSNs

Done

Lines 212-212. Does this section refer to a figure? Its rational is hard to follow otherwise.

It was mistakenly added to the MS, now omitted.

Lines 217-219 This is hard to understand. It might be more appropriate at the very end of this section (say line 224) because it then makes more sense.

Done. The whole Methods section was re-arranged, more explanations added to the figures (+1 new figure).

Table 1. Define AFs.

Removed AF from the table, it caused confusion.

Figure 3 caption. Relative sensitivity is plotted radially in ? dB steps

Done, added explanation:

C, E. Relative sensitivity (inverse of relative threshold) is plotted radially in 2 dB steps, connected by thick black line. D, F. Absolute sensitivity (inverse of absolute threshold) is plotted radially in 3 (D) and 4 (F) dB SPVL steps, connected by thick black line.

Figure 4 caption. It would be helpful to have vertical scale bars for A, B, F and G

Done.

Line 410 “Ratios between individual frequencies in pairs and triplets of units”. The significance of these measurements is not evident. Please provide a brief rational for making these measurements.

This indeed did not relate to the aim of our study, but we found it worthwhile to present such analysis of our large dataset of individual frequencies. The fact that the ratios between AFs were distributed as the integer ratios suggests that paired units (we believe that they are the PSNs belonging to the same scolopidium) do interact, and such interaction may stabilize the individual tuning frequencies. This result also nicely fits the analysis of auditory interactions in mosquitoes by Aldersley et at 2016. Added the following paragraph to Results:

Although this did not relate to the main scope of this study, after obtaining a siginificant dataset (n=240) of individual frequencies (AFs) of the JO auditory units, most of which were measured pairwise within the same recording sites, we analyzed the frequency ratio in these pairs. A regular non-random nature of the ratio would speak in favor of interactions between the paired units, also confirming our previous findings in male mosquitoes [13].

Figure 5 caption. Vertical scales in Fig 5E-G would be helpful.

Done.

Lines 435-426. Land discovered this much earlier. M. F. Land, Head movement of flies during visually guided flight. Nature 243, 299–300 (1973).

Thank you, we replaced the link.

Line 458. By “ones” do you mean JO sensory units?

Yes. Changed to "units"

Reviewer 2 Report

There is a wealth of useful data in this paper but it needs to be communicated such that someone who hasn't carried out the experiments can understand. I really struggled to understand what was going on as I read through the manuscript. 

The antennae orientation was very nice (very nice figure 1). I have an issue with the claim that individual "units" are being recorded. Is the claim that individual units are axons? Without intracellular recordings I highly doubt this. I guess one way that this can be tested (presumably with data that you have already collected) is to measure the response at twice the stimulation frequency when the speaker is stimulating in line to the flagellum. If you are recording individual units the 2f component should be very low. If you are recording from the whole JO I would expect 2f to be large (about half the amplitude of the 1f response when the speaker is perpendicular). This would really help to settle the locality of your recordings.

I've made comments throughout the manuscript up to page 11 when I lost understanding too far.

The quality of English requires improvement but the larger issue is the explanation and communication .of the science

Author Response

Thank you for clear and helpful review of our paper. Our responses are copied below, however, they are more conveniently presented in the attached pdf file. The Reviewer's comments formatted in Italic.

I have an issue with the claim that individual "units" are being recorded. Is the claim that individual units are axons? Without intracellular recordings I highly doubt this.

We agree that, using extracellular recording, like in this study, we record from a number of axons of the JO sensory neurons. However, we have reasons to believe that this number is not large. How can we disprove the hypothesis that we did not record anything more than the combined activity of the JO?

We believe that we have already done it in our previous studies:

Lapshin Vorontsov 2016 Frequency organization of the Johnston’s organ in male mosquitoes (Diptera, Culicidae) Journal of Experimental Biology (2017) 220, 3927-3938 doi:10.1242/jeb.152017

Lapshin, D.N.; Vorontsov, D.D. Frequency tuning of individual auditory receptors in female mosquitoes (Diptera, Culicidae). J. Insect Physiol. 2013, 59, 828–839, doi:10.1016/j.jinsphys.2013.05.010.

Here are the main points from these studies.

- Intracellular recordings from the axons of the antennal nerve. They are not easy to maintain for a significant time interval, as the movements of non-narcotized mosquito affects the stability of recording (while narcosis changes the functioning of the JO). However, we managed to obtain such recordings.

- Effects of mechanical shift of the recording electrode: this procedure often caused an abrupt change from one AF to another, which would not happen if we recorded a compound response

from a large pool of axons. We even performed a back and forth switch between different recording sites (showing different and distinct AFs at each site). The same was true for the auditory thresholds (measured without the feedback stimulation): slight axial shifts of the electrode significantly changed the frequency preference and/or absolute thresholds of response.

Also from the results of this study: the combined activity of the JO would not demonstrate such clear (and different) directional diagrams (like the ones shown in our Figure 3). If there were many differently oriented neurons recorded simultaneously, their summary directional diagram would be more or less circular, with no difference in thresholds of 14 dB or more at different directions. Also, the positive feedback stimulation, which by its principle (as we see it) gave preference to the axons located closer to the electrode tip (they took preference by choosing the frequency of autoexcitation), demonstrated similar directional properties at the same recording sites (we provided the comparison of such data in Fig.4), thus confirming the locality of recording.

We did not include neither intracellular recordings not other tests of locality of recordings into this study for the following reasons. 1) We believed the issue was already solved and the relevant results published. 2) The current study already contained complex and not easy-to-understand data, thus we tried to minimize the inclusion of already published data. 3) In the current study we used only extracellular responses, as they are much more stable and allow to make lengthy measurements.

As we cannot say exactly how many axons contributed to each recording, here we use the

terms ‘unit’ or ‘sensory unit’ in the sense of one or several axons belonging to the PSNs of the JO, closely located within the antennal nerve and sharing indistinguishable frequency and phasic properties, thus representing a single functional unit.

Specific distribution (Fig.6FG) of ratios between a very limited (2 to 3) number of AFs recorded at each site also indirectly suggests that the number of units contributing to the recording was low. If we recorded the summary signal from the whole JO, or even from a large proportion of axons in the antennal nerve, the AFs and their ratios (as well as the directional characteristics) would be more or less the same in each recording.

The method of positive feedback stimulation allowed us, as we believe, to make a functional sharpening of the electrode and to record from a single axon at a time. However, as the method itself raised significant discussion before, to which we contributed by publishing different controls for possible artifacts, here we did not press on that we recorded from single PSNs. For this study it was only important that the recording was focal (from a limited number of axons rather than from the whole JO). Also, we did not rely on feedback stimulation in measurements of auditory thresholds.

I think you need to be clear what you mean by units. I doubt that you are detecting individual axons but at best a sub-sample.

We added an explanation of what we mean by unit to the methods:

In this study, a number of axons of primary sensory neurons (PSNs) contributed to the extracellular recording. Although we could not estimate this number in each recording, our previous studies [21, 23], which included the intracellular recordings from the axons of PSNs, suggested that the number of simultaneously recorded axons was not large. In our experience, the stability of intracellular recording from the JO axons is hard to maintain during the time required for the measurement of thresholds at different frequencies or directions of sound. For the sake of stability of recording, in this study we used only extracellular recording of neuronal responses. We use the terms ‘unit’ or ‘sensory unit’ in the sense of one or several axons belonging to the PSNs of the JO, closely located within the antennal nerve and sharing indistinguishable frequency and phasic properties, thus representing a single functional unit.

If you are recording from only a few "units" when you stimulate along the same axis of the flagellum you should get no response at 2f of the stimulus frequency. If this s the case you need to show this data because I am not convinced that you are recording anything more than the combined activity of the JO

According to our understanding, the 2f component is a feature of the extracellular recording from the JO and not the result of combination of signals from the opposite prongs or of any of such mechanisms, involving the geometry of the JO. We did not see the substantial level of the 2f component in the intracellular recordings from the axon, but 2f was clearly present in the same recording just before going intracellular (the same was observed in other similar recordings). Here we attach the Fig.2 of Lapshin & Vorontsov (2016) – the 2f component disappeared between the fragments of recording 1 and 2.

According to this, we see no rationale in the test with axial (along the flagellum) stimulation to prove or disprove the locality of recording.

'polar patterns'

this is a very cryptic way to name it. Why not just call is autoexcitation frequnecy?

I dont really understand what you mean by directional characteristics or what was mesaured here. Was it a threshold at the AF?

We are sorry that we did not make enough explanations of our method. This could be one of the reasons for difficulty in understanding the results.

Polar pattern is not an autoexcitation frequency. It is a function of autoexcitation threshold against the direction of stimulation. Similarly, the directional characteristic is a similar function of absolute auditory thresholds against the direction of sinusoidal stimulation. Both polar pattern and directional characteristic are based on a number of measurements repeated each 15°. However, these functions are different – a polar pattern is unilateral while a directional characteristic is bilateral. We searched for appropriate terms to name these functions and found them in radioelectronics and microphone engineering but, unfortunately, not in physiology.

We tried to explain the design of experiments exactly where the Reviewer's questions appeared:

Two kinds of directional measurements were performed, depending on the type of stimulation. 1. The thresholds of feedback stimulation were measured depending on the direction of a sound wave, producing unipolar plot (hereafter 'polar pattern', example in Figure 4C). 2. Using the measured AF value as the frequency of sinusoidal stimulation, the absolute auditory thresholds were measured depending on the direction of a sound wave, producing a bipolar plot (hereafter 'directional characteristics', example in Figure 4D).

Also, we re-ordered paragraphs in the Methods, as suggested by the Reviewer, and added a separate figure/table to explain the course of experiment and the kinds of data obtained at each stage. We hope that it will help readers to understand our work.

I think a figure would really help to explain your experimental setup

We added two panels to Fig.2 that would explain the idea of a vector superposition of acoustic waves at the point of mosquito's antenna.

what areas do the quandrants correspond to? maybe a figure in the methology section would help.

We provided the explanation of quadrants in the figure where they first appear, now Fig. 5 (the figure numbers changed due to addition of one figure after Fig.2)

so I asume 0 degrees is the midline?

Could be wroth overlaying a mosquito on one of the polar plots to make it clear

Overlaying a mosquito on a plot could be misleading, as the polar coordinate system that we use is centered at the antenna and not at the mosquito body (at the later stage of our data presentation we come to the auditory space around a mosquito, Results, 3.6). At the same time, it is not easy to show how an antenna looks like when viewed along its flagellum from the head of mosquito. In the figure with examples of single polar patterns we added a photo of a mosquito head, with the polar coordinate system overlaid over it (the angle in that figure is counted counter-clockwise). We hope that it became clear now.

did you calibrate the speakers to keep their frequency resonse liniear across frequencies. If not, maybe limit your findings to where the speaker remains linear?

Of course, the speakers were calibrated in the whole frequency range used in this study. As there were more questions about the method from other Reviewers, we added the detailed description of the stimulation system to the Methods (however, it almost copies the one published in Lapshin & Vorontsov 2019 JEB [13]):

We used two orthogonally oriented Scandinavia 75 (DLS, Sweden) stationary speakers to create a vector superposition of acoustic waves at the point of mosquito antenna (Fig. 2AB), as described in detail in [13]. The mosquito was positioned at the crossing of the axes of two speakers in such a way that the antenna′s flagellum was perpendicular to the directions of sound waves originating from each of the two speakers (Fig.2A). This approach enabled to set the desired direction of the acoustic vector relative to the antenna flagellum.

The moving parts of the speakers had a low resonant frequency (90 Hz). Due to the considerable response lag of the dome of the speaker and its support, the emission phase delay increased with the signal frequency up to the point of inversion. To stabilize the phase delay, a phase correction depending on the stimulation frequency was included in the speaker control circuit. Both speakers were covered by metallic mesh to screen the recording electrode from the electrical signals that drove the speakers.

The sinusoidal stimuli were generated by the digital-to-analog converter LA-DACn10m1 (Rudnev-Shilyaev, Russian Federation). Acoustic calibration of the stimulating device was performed with an NR-231-58-000 differential capacitor microphone (Knowles Electronics, USA) attached to a micropositioner with axial rotation feature and set in the position of the mosquito. The same microphone put in 2 cm from the mosquito was used to record the stimulation signals during the experiments.

The speakers were powered from the home-made amplifier via a passive Sin–Cos (SC) transducer which produced two derived signals with the amplitudes,

where A1 and A2 are the amplitudes of the signals for the first and the second speaker, respectively; U is the alternating voltage at the input of the SC transducer; φ is the angle between the dorso-ventral axis passing through the mosquito′s head and the vector of vibration velocity of air particles. An increase in φ corresponds to clockwise rotation of the velocity vector when viewed from the mosquito′s head along the antenna (Fig.2B).

The resulting direction of the air vibration velocity in the stimulating system was determined by the vector superposition of the signals from both speakers. Changes in the sound wave direction relative to the mosquito in 15° steps were accomplished by coordinated switching of voltage dividers in the SC transducer. For those angles at which the values of the functions sin(φ+45) or cos(φ+45) were negative, the signal polarity was inverted by switching the terminals of the speakers.

The differential microphone together with its amplifier was previously calibrated in the far field using the B&K 2253 sound level meter with a B&K 4135 microphone (Brüel & Kjær, Denmark). All sound level data in this study are given in the logarithmic scale in dB RMS SPVL (root mean square sound particle velocity level), with a reference level of 0 dB being equal to 4.85×10‑5 mm/s, which corresponds in the far field to the standard reference sound pressure of 20 µPa.

We also added a detailed description of the positive feedback stimulation to the Methods.

I think you can come to a larger conclusion then what you have. There is ongoing research to use acoustic traps for males. If we can crack the prupose of female hearing this might be a game-changer for such apporahces to control mosqitoes.

We would prefer to be on a safe side in our conclusions. This study is indeed a step towards our understanding of the purpose of female hearing, but the results only indirectly answer the question. However, we added to Discussion a subsection "On the possible functions of hearing in female mosquitoes".

All other comments through the text were gratefully accepted.

Reviewer 3 Report

This is a very interesting and complex manuscript that contributes to advance the field of mosquito auditory perception. Although the paper is very interesting and relevant, I found it very difficult to follow. In the first instance, I would suggest that the authors try to facilitate the understanding of the manuscript by improving the accessibility of the methodology section, improving the diagrams and its labelling to illustrate the concepts explained in the methods section. It seems to me that the authors base a lot of concepts in their previous work, which is understandable, but it should be possible to understand this manuscript independently. I refer to specific sections in the comments below. Moreover, the discussion section in very difficult to follow because the authors discuss figures and concepts that have not been previously introduced in the Results. It has been almost impossible for me to follow the discussion after a few hours dedicated to this review. I would ask the authors to improve these sections before a more thorough review can be done of the Discussion. Also, abbreviations (e.g. PSN neurons or AF) need to be introduced.

Methods

In general, along the methods, could the authors clearly state which parameters were extracted from the different methodologies used? Currently, it requires a lot of effort from the reader to relate the written methods to the plots shown in the results sections. 

Line 155: Where the lateral and horizontal images acquired simultaneously or were these different measurements? Where the taken from same or different mosquitoes? This part of the methods needs a bit more detail. Also, the authors do not mention the circadian time at which the measurements were taken, this would be highly relevant for this type of analysis. Could the authors comment on this? Also, no information is provided regarding the mating status of the females, was this investigated, checked?

Line 162: It is not clear the angles measured by the authors from the photographs. In the methods, it is written that they also used “head to horizon”. However, in Fig. 1 this is not shown, could the authors please clarify? Also, could the authors provide (even in supplementary material), an example of how the raw images look like? Was it always possible to unequivocally distinguish the antenna? Did the authors apply some filters in Fiji to visualize the images? Also, I assume that the mosquitoes were glued to avoid movements of the head or body parts, can the authors include this information?

Figure 2: This figure needs a better annotation. Could the authors label the sonogram and the different frequency responses? e.g: “Note the two traces of simultaneous excitation at different frequen- 200 cies (69 and 114 Hz). ” Do these correspond to “tonal pulses” that the authors apply during the searching procedure?

Line 202-212: This paragraph seems to combine “Methods” and “results”. Also, it is confusing because the values included (for threshold, excitation…) do not agree with Fig. 2B, could the authors just include here the relevant methods, and be consistent between the values shown in the methods and the Figure (to facilitate the understanding of the reader)? Also, could the authors provide a diagram that shows how the direction of the sound wave is measured (is it in relation to the mosquito head) just by drawing the speakers in relation to the mosquito? This would greatly facilitate the understanding of the methods followed. Also, could the authors clearly state which parameters are extracted from these measurements to generate the plots in the manuscript?

Line 211: Which sonogram are the authors referring to? Fig. 2 does not seem to represent this data…

Line 218: These 162 experiments belong to 162 mosquitoes? Please, clarify.

Line 219: What does “individual characteristics” mean?

Line 233-242 (Fig.3): Could the authors include the position of speakers in the figure? It is not clear to me why the authors use two speakers instead of a single one, could the authors explain this?

Figure 3: This figure needs a better annotation. What do the dark lines represent? Is it Ai? Which thresholds are shown in the plots (relative or absolute)? In D, which represent the same sensory unit as in C, why do the authors use a sinusoid of 100 Hz instead of the tuning frequency as defined in C? Also, could the authors explain in Fig. 2B (sonogram) and lines 202-211, which values are used for stimulation in “positive feedback” and “sinusoidal” experiments? Could the authors define “absolute thresholds”?

Line 313: AF meaning is introduced here, but it has been used before. Please, include this information the first time AF is used. 

Line 324: This correction for the temperature seems rather simplistic to me, could the authors comment on this? What about non-linear responses of JO to changes in temperature? 

Results

Could the authors define sensory units? How is the anatomical organization of the sensory units in the JO?

Line 340. Here the authors say “typical threshold directional characteristic of an individual sensory unit measured 339 with the sinusoidal pulse stimulation was symmetrically bi-directional, showing a classic 340 figure-eight pattern (Fig.3BD). ” However, in the figure they indicate that Fig. 3D represents two pair of units. Can they please clarify?

Discussion:

Please, more work needs to be dedicated to the discussion. The authors introduce findings that should be included in the Results. 

It is not clear to me why authors have included all information regarding Fig. 6 in the Discussion. This Figure should be introduced and explained in the result section. 

Equation 3 should be introduced in the methods if used in the calculations. 

Could the authors discuss the relevance of their results from a sensory ecological perspective?

Author Response

Thank you for clear and helpful review of our paper. Our responses are copied below, however, they are more conveniently presented in the attached pdf file. The Reviewer's comments formatted in Italic.

This is a very interesting and complex manuscript that contributes to advance the field of mosquito auditory perception. Although the paper is very interesting and relevant, I found it very difficult to follow. In the first instance, I would suggest that the authors try to facilitate the understanding of the manuscript by improving the accessibility of the methodology section, improving the diagrams and its labelling to illustrate the concepts explained in the methods section. It seems to me that the authors base a lot of concepts in their previous work, which is understandable, but it should be possible to understand this manuscript independently. I refer to specific sections in the comments below. Moreover, the discussion section in very difficult to follow because the authors discuss figures and concepts that have not been previously introduced in the Results. It has been almost impossible for me to follow the discussion after a few hours dedicated to this review. I would ask the authors to improve these sections before a more thorough review can be done of the Discussion. Also, abbreviations (e.g. PSN neurons or AF) need to be introduced.

We are grateful to the Reviewer for spending much time trying to understand our study. We are sorry that the presentation of our results was not clear enough. In attempt to improve the paper we added the detailed descriptions of the directional acoustic stimulation and the feedback stimulation methods (which were published elsewhere), extended the existing methodical figure, added a new figure explaining the experimental design, and re-arranged the Discussion.

Methods

In general, along the methods, could the authors clearly state which parameters were extracted from the different methodologies used? Currently, it requires a lot of effort from the reader to relate the written methods to the plots shown in the results sections.

We added a new figure (now Fig.3) that explains the consequence of electrophysiological measurements taken for each sensory unit, with references to the sub-sections in the Methods and to the figures that show results obtained at each stage of experiment. Hopefully this will ease the understanding of the methods and results.

Line 155: Where the lateral and horizontal images acquired simultaneously or were these different measurements? Where the taken from same or different mosquitoes? This part of the methods needs a bit more detail. Also, the authors do not mention the circadian time at which the measurements were taken, this would be highly relevant for this type of analysis. Could the authors comment on this? Also, no information is provided regarding the mating status of the females, was this investigated, checked?

The following sentences, answering the above questions, were added to the Methods.

Lateral and dorsal imaging was for the same mosquito was done sequentially, one series after another. Experiments were performed at the end of August, from 1 p.m. to 6 p.m., substantially before the swarming hours of Culex pipiens. The mating status of the female mosquitoes was not checked.

Line 162: It is not clear the angles measured by the authors from the photographs. In the methods, it is written that they also used “head to horizon”. However, in Fig. 1 this is not shown, could the authors please clarify?

We did not include "head to horizon" parameter in the final analysis. Now excluded it from the Methods.

Also, could the authors provide (even in supplementary material), an example of how the raw images look like? Was it always possible to unequivocally distinguish the antenna?

We added several raw images to the Supplementary.

From the large number of shots taken, for the study we selected only those where 1) the antennae could be easily distinguished 2) the antennae were in correct orientation relative to the plane of view and 3) where a mosquito was further than 1cm from the walls of the container. We added examples of images both suitable and unsuitable for analysis (commented them accordingly).

Did the authors apply some filters in Fiji to visualize the images?

We did not apply any filters to the images, only cropped them before measuring the angles in FIJI (added examples of cropped images to the Supplementary).

The selected set of images (3 per each mosquito) was cropped and then the following angles were manually measured using the FIJI...

Also, I assume that the mosquitoes were glued to avoid movements of the head or body parts, can the authors include this information?

The flying mosquitoes were not glued, of course. In the electrophysiological experiment, they were glued. We cite our previous work where the whole electrophysiological method was described in detail: Lapshin, D.N.; Vorontsov, D.D. Directional and frequency characteristics of auditory neurons in Culex male mosquitoes. J. Exp. Biol. 2019, 222, doi:10.1242/jeb.208785.

In short, "... mosquitoes were glued to a copper-covered triangular plate with a flour paste with 0.15 M sodium chloride added. … The head of the mosquito was glued to its body by a bead of nail varnish (partially dried in advance to minimize the drying time). The mosquito could still move its antennas, but this was visually controlled."

Figure 2: This figure needs a better annotation. Could the authors label the sonogram and the different frequency responses? e.g: “Note the two traces of simultaneous excitation at different frequen- 200 cies (69 and 114 Hz). ” Do these correspond to “tonal pulses” that the authors apply during the searching procedure?

We extended the figure caption:

D. Sonogram (frequency spectrum of signal against time) of response to a positive feedback stimulation, when the electrical potential recorded from the antennal nerve is amplified and fed to the speaker in real time. Colour represents the relative amplitude of response, from blue to red. The recording starts in silence (–17 dB below the threshold), then the level of feedback is gradually increased. The traces of the feedback effects appear at –9 dB (narrow-band frequency-selective increase of noise). At 0 dB (threshold) appears a sustained autoexcitation at 114 Hz and a weaker autoexcitation at 69 Hz. Two traces of simultaneous excitation at different frequencies (69 and 114 Hz) represent differently tuned sensory units. Traces at higher frequencies are harmonics.

Also, could the authors provide a diagram that shows how the direction of the sound wave is measured (is it in relation to the mosquito head) just by drawing the speakers in relation to the mosquito? This would greatly facilitate the understanding of the methods followed. Also, could the authors clearly state which parameters are extracted from these measurements to generate the plots in the manuscript?

We extended Fig.2 by adding the diagrams with mosquito head and the speakers. We added a new figure (now Fig.3) to explain the sequence and the results of measurements, and mentioned the parameters extracted at each stage. We also extended Fig.4 by showing the polar coordinate system over the photo of a mosquito head.

Line 202-212: This paragraph seems to combine “Methods” and “results”. Also, it is confusing because the values included (for threshold, excitation…) do not agree with Fig. 2B, could the authors just include here the relevant methods, and be consistent between the values shown in the methods and the Figure (to facilitate the understanding of the reader)?

Line 211: Which sonogram are the authors referring to? Fig. 2 does not seem to represent this data…

Also, could the authors explain in Fig. 2B (sonogram) and lines 202-211, which values are used for stimulation in “positive feedback” and “sinusoidal” experiments?

We are sorry, this paragraph appeared in the MS by mistake. It did not relate to Fig.2. Now completely omitted. For Fig.2, the parameters of stimulation are now clearly stated in its caption.

Line 218: These 162 experiments belong to 162 mosquitoes? Please, clarify.

Line 219: What does “individual characteristics” mean?

Clarified in the text and replaced "individual characteristics" by "polar patterns":

In 162 mosquitoes studied, of which in 24 the auditory responses were recorded from the right JO, totally 290 (50 from the right JO) polar patters were obtained (Table 1)

Line 233-242 (Fig.3): Could the authors include the position of speakers in the figure? It is not clear to me why the authors use two speakers instead of a single one, could the authors explain this?

Before we relied on the previously published detailed description of our method. Now we significantly extended the Methods section, describing the acoustic stimulation system:

We used two orthogonally oriented Scandinavia 75 (DLS, Sweden) stationary speakers to create a vector superposition of acoustic waves at the point of mosquito antenna (Fig. 2AB), as described in detail in [13]. The mosquito was positioned at the crossing of the axes of two speakers in such a way that the antenna′s flagellum was perpendicular to the directions of sound waves originating from each of the two speakers (Fig.2A). This approach enabled to set the desired direction of the acoustic vector relative to the antenna flagellum.

The moving parts of the speakers had a low resonant frequency (90 Hz). Due to the considerable response lag of the dome of the speaker and its support, the emission phase delay increased with the signal frequency up to the point of inversion. To stabilize the phase delay, a phase correction depending on the stimulation frequency was included in the speaker control circuit. Both speakers were covered by metallic mesh to screen the recording electrode from the electrical signals that drove the speakers.

The sinusoidal stimuli were generated by the digital-to-analog converter LA-DACn10m1 (Rudnev-Shilyaev, Russian Federation). Acoustic calibration of the stimulating device was performed with an NR-231-58-000 differential capacitor microphone (Knowles Electronics, USA) attached to a micropositioner with axial rotation feature and set in the position of the mosquito. The same microphone put in 2 cm from the mosquito was used to record the stimulation signals during the experiments.

The speakers were powered from the home-made amplifier via a passive Sin–Cos (SC) transducer which produced two derived signals with the amplitudes,

where A1 and A2 are the amplitudes of the signals for the first and the second speaker, respectively; U is the alternating voltage at the input of the SC transducer; φ is the angle between the dorso-ventral axis passing through the mosquito′s head and the vector of vibration velocity of air particles. An increase in φ corresponds to clockwise rotation of the velocity vector when viewed from the mosquito′s head along the antenna (Fig.2B).

The resulting direction of the air vibration velocity in the stimulating system was determined by the vector superposition of the signals from both speakers. Changes in the sound wave direction relative to the mosquito in 15° steps were accomplished by coordinated switching of voltage dividers in the SC transducer. For those angles at which the values of the functions sin(φ+45) or cos(φ+45) were negative, the signal polarity was inverted by switching the terminals of the speakers.

Figure 3: This figure needs a better annotation. What do the dark lines represent? Is it Ai? Which thresholds are shown in the plots (relative or absolute)?

We extended annotation of this figure:

A. The polar coordinate system centered at the left antenna, mosquito is viewed posteriorly. The left antenna is hidden behind a mosquito body and is oriented perpendicular to the image plane. Red arrow shows the positive (clock-wise) direction of angular axis used in all polar diagrams for both left and right JO. B. Photo of a mosquito head shows the same coordinate system, centered at the left antenna. C. Polar pattern of a single responding unit (AF 112 Hz), consisting of one unipolar petal. D. Directional characteristic of the same unit measured by sinusoidal stimulation, containing two almost symmetrical petals (measured independently); threshold of response 32 dB SPVL at 112 Hz. E. Polar patterns of two units recorded at the same site and responding in anti-phase; the tuning frequencies are 104 Hz (unit 1) and 77 Hz (unit 2); F. Directional characteristic of the same pair of units as in E, measured as a threshold to 100 Hz sinusoidal stimulation. C, E. Relative sensitivity (inverse of relative threshold) is plotted radially in 2 dB steps, connected by thick black line. D, F. Absolute sensitivity (inverse of absolute threshold) is plotted radially in 3 (D) and 4 (F) dB SPVL steps, connected by thick black line.

In D, which represent the same sensory unit as in C, why do the authors use a sinusoid of 100 Hz instead of the tuning frequency as defined in C?

In this case we used the stimulation frequency (100 Hz) from the interval between the two AFs (77 and 104 Hz) in order to demonstrate responses from both units.

Could the authors define “absolute thresholds”?

Added the definition to the Methods:

Such auditory thresholds, measured in response to sinusoidal stimulation and expressed in dB RMS SPVL, are hereafter called 'absolute thresholds' or just 'thresholds'.

Line 313: AF meaning is introduced here, but it has been used before. Please, include this information the first time AF is used.

Done, but other Reviewers required to explain the abbreviations separately in other places of the MS (as we understood it).

Line 324: This correction for the temperature seems rather simplistic to me, could the authors comment on this? What about non-linear responses of JO to changes in temperature?

To our knowledge, there is no data on the non-linear responses of JO to changes in temperature. It is known that the frequency tuning of the JO, as well as the wingbeat frequency, highly depends on the ambient temperature (Costello, 1974). In previous studies, we initially used the coefficient of 2% per 1°C, based on the study of Villarreal et al., (2017). Then, based on our own behavioral data (Lapshin Vorontsov 2021), we established the presently used equation. In the majority of mosquito studies, no such correction is used at all, so this simplistic correction is better than nothing.

Results

Could the authors define sensory units? How is the anatomical organization of the sensory units in the JO?

We added definition of 'units' to the Methods:

In this study, a number of axons of primary sensory neurons (PSNs) contributed to the extracellular recording. Although we could not estimate this number in each recording, our previous studies [21, 23], which included the intracellular recordings from the axons of PSNs, suggested that the number of simultaneously recorded axons was not large. In our experience, the stability of intracellular recording from the JO axons is hard to maintain during the time required for the measurement of thresholds at different frequencies or directions of sound. For the sake of stability of recording, in this study we used only extracellular recording of neuronal responses. We use the terms ‘unit’ or ‘sensory unit’ in the sense of one or several axons belonging to the PSNs of the JO, closely located within the antennal nerve and sharing indistinguishable frequency and phasic properties, thus representing a single functional unit.

In the mosquito JO, the sensory neurons are grouped into pairs or triplets, forming sensillae, or scolopidia. We have reasons to believe that the paired responses that we record in the antennal nerve are produced by the neurons belonging to the same sensilla. One of the indirect evidences in favor of this view is presented by non-random ratios of frequencies (Fig.6FG) together with the opposite (antiphase) directional properties in such paired responses. However, this is a speculation and we do not press on it.

Line 340. Here the authors say “typical threshold directional characteristic of an individual sensory unit measured 339 with the sinusoidal pulse stimulation was symmetrically bi-directional, showing a classic 340 figure-eight pattern (Fig.3BD)." However, in the figure they indicate that Fig. 3D represents two pair of units. Can they please clarify?

It was a mistake, we meant only Fig.3B. Corrected.

Discussion:

Please, more work needs to be dedicated to the discussion. The authors introduce findings that should be included in the Results.

It is not clear to me why authors have included all information regarding Fig. 6 in the Discussion. This Figure should be introduced and explained in the result section.

Equation 3 should be introduced in the methods if used in the calculations.

We used this figure (Fig. 6 in the original version of the MS, now Fig.7) to aid our interpretation of the experimental data. Such interpretations, as we believed, did not relate to Results. Assuming this was our mistake and that all figures and equations in a paper must formally be introduced in Results, we moved to Results the sub-sections 'Directionality of sensory units in three dimensions' and most of 'Equisignal zones', which introduce and explain several equations and Figure 6 (in the original numbering, now Figure 7).

We also re-arranged other sub-sections in Discussion. Hopefully they became easier to read now.

Could the authors discuss the relevance of their results from a sensory ecological perspective?

We added to Discussion a subsection "On the possible functions of hearing in female mosquitoes".

Round 2

Reviewer 2 Report

The revised figures help greatly with the understanding. The English could still be improved (see comments on the attached pdf) but is not essential for publication.

Author Response

Thank you very much for helping us with the English style.

We accepted most of your suggestions. In several cases, however, we preferred not to shorten the text, but to rephrase it: the current version of MS is a result of multiple revisions, and we often had to add explanations for things that seemed trivial to ourselves. Most probably, such misunderstandings were mostly due to our inexperience in English writing. We will try to remember to put the verb closer to the beginning of the sentence.

As for the physiological asymmetry of the mosquito JO, we tried to say that currently it is the only indication of binaural mechanisms available to us. Methodically, it is not an easy task to study how the signals from two JOs are combined in the mosquito brain. However, even the available indirect data can give a future researcher an insight of what to search for.

We also found a mistake in our calculation of inter-antennal angle, however, it did not affect our conclusions. In fact, the fit between the predicted value of the upward inclination of the antennal plane and its measured value became even better (it was 38.6° vs 35.5±6.7° and after correction of a mistake became 35.7° vs 35.5±6.7°).

Reviewer 3 Report

Dear authors,

Thanks for significantly improving the clarity of the manuscript. I have now really enjoyed reading it, following your smart approaches to disentangle the mechanisms of mosquito auditory perception. I think the methods and results are much clear now, and therefore I recommend that this paper is published.

I just would suggest to the authors that they include in their references the following article that very much relates to the mosquito sensory perception in the swarm and to the use of distortion products for mosquito acoustic perception:

Somers J, Georgiades M, Su MP, Bagi J, Andrés M, Alampounti A, Mills G, Ntabaliba W, Moore S, Spaccapelo R, Albert J. (2022) Hitting the right note at the right time: Circadian control of audibility in Anopheles mosquito mating swarms is mediated by flight tones. Science Advances 6 (2): eabl4844. doi: 10.1126/sciadv.abl4844.

Author Response

Thank you very much for spending your time on reading our manuscript and, again, for your previous questions and comments, which allowed us to make the text clearer.

As for the study of Somers et al (2022) - surely we know that study, it was cited in the original version of our MS, published as a preprint, where we reviewed the findings on the harmonic convergence and the use of distortion products for auditory perception in mosquitoes. However, since then we received a number of review responses, some of which suggested to remove the lengthy passage as unrelated to the topic of our study.

We included the citation of Somers et al (2022) now.

We also found a mistake in our calculation of inter-antennal angle, however, it did not affect our conclusions. In fact, the fit between the predicted value of the upward inclination of the antennal plane and its measured value became even better (it was 38.6° vs 35.5±6.7° and after correction of the mistake became 35.7° vs 35.5±6.7°).